# Integrative proteomic and phosphoproteomic profiling of prostate cell lines

**Maria Katsogiannou**[1,2�], **Jean-Baptiste Boyer**[1☉], **Alberto Valdeolivas**[3,4,5☉], **Elisabeth Remy**[3], **Laurence Calzone**[6], **Stéphane Audebert**[1], **Palma Rocchi**[1‡*], **Luc Camoin**[1‡*], **Anaïs Baudot**[3,4‡*]

**1** Aix Marseille Univ, CNRS, INSERM, Institut Paoli-Calmettes, CRCM, Marseille, France, **2** Obstetrics and Gynecology department, Hôpital Saint Joseph, Marseille, France, **3** Aix Marseille Univ, CNRS, Centrale Marseille, I2M, Marseille, France, **4** Aix Marseille Univ, INSERM, MMG, Marseille, France, **5** ProGeLife, Marseille, France, **6** Mines Paris Tech, Institut Curie, PSL Research University, Paris, France

☉ These authors contributed equally to this work.
‡ These authors also contributed equally to this work.
* palma.rocchi@inserm.fr (PR); luc.camoin@inserm.fr (LC); anais.baudot@univ-amu.fr (AB)

**Data Availability Statement:** The mass spectrometry proteomics data, including search, are available at the ProteomeXchange Consortium (www.proteomexchange.org) via the PRIDE

## Abstract

### Background

Prostate cancer is a major public health issue, mainly because patients relapse after androgen deprivation therapy. Proteomic strategies, aiming to reflect the functional activity of cells, are nowadays among the leading approaches to tackle the challenges not only of better diagnosis, but also of unraveling mechanistic details related to disease etiology and progression.

### Methods

We conducted here a large SILAC-based Mass Spectrometry experiment to map the proteomes and phosphoproteomes of four widely used prostate cell lines, namely PNT1A, LNCaP, DU145 and PC3, representative of different cancerous and hormonal status.

### Results

We identified more than 3000 proteins and phosphosites, from which we quantified more than 1000 proteins and 500 phosphosites after stringent filtering. Extensive exploration of this proteomics and phosphoproteomics dataset allowed characterizing housekeeping as well as cell-line specific proteins, phosphosites and functional features of each cell line. In addition, by comparing the sensitive and resistant cell lines, we identified protein and phosphosites differentially expressed in the resistance context. Further data integration in a molecular network highlighted the differentially expressed pathways, in particular migration and invasion, RNA splicing, DNA damage repair response and transcription regulation.

partner repository with datasets identifiers PXD004970 and PXD004992.

**Funding:** This work was supported by ProGelife and the French "Plan Cancer 2009-2013" (Systems Biology call). The funders had no role in study design, data collection and analysis, decision to publish, or preparation of the manuscript.

**Competing interests:** This work was supported by the French Plan Cancer 2009-2013 (Systems Biology call). The funders had no role in study design, data collection and analysis, decision to publish, or preparation of the manuscript. The funder ProGeLife provided support in the form of salaries for author AV, but had no role in the study design, data collection and analysis, decision to publish, or preparation of the manuscript. This commercial affiliation does not alter our adherence to PLOS ONE policies on sharing data and materials. The specific roles of the authors are articulated in the 'author contributions' section.

## Conclusions

Overall, this study proposes a valuable resource toward the characterization of proteome and phosphoproteome of four widely used prostate cell lines and reveals candidates to be involved in prostate cancer progression for further experimental validation.

## Introduction

Prostate cancer (PC) is a major public health issue in industrialized countries, mainly because patients relapse by castration-resistant (CR) disease after androgen deprivation [1, 2]. PC is associated to a panel of clinical states characterized by tumor growth, hormonal status (castration-sensitive (CS) or CR) and presence/absence of metastases. After androgen deprivation therapy, the disease usually progresses to castration-resistant prostate cancer (CRPC), which is highly aggressive and incurable, and jeopardizes the patient's lifespan and quality of life. This progression involves several molecular mechanisms such as ligand-independent androgen receptor (AR) activation, AR expression loss or adaptive upregulation of anti-apoptotic genes (for review [3]).

Despite an existing treatment guideline for PC and novel clinical trials for CRPC [4, 5], major challenges remain to understand and treat these cancers appropriately. Large-scale -omics approaches, able to monitor cancer-induced changes at the cellular level, are among the most promising strategies. Proteomic strategies, by measuring the abundance and activity of proteins, have the ability to directly reflect the functional activity of cells, and to point to deregulations in the most druggable cellular components. In this context, several proteomic studies started to map the landscape of the PC proteome [6–10]. These studies identified biomarkers, such as the proneuropeptide $Y^7$, as well as proteomic changes associated to PC progression (e.g., increased anabolic processes and oxidative phosphorylation in primary PC as described by [7] and [8]). Overall, such analyses are valuable not only for diagnosis, but also for providing mechanistic details related to disease etiology and progression.

These proteomic approaches focused on protein quantification, but neglect protein phosphorylation, a key point in the measurement of cellular activity. Protein phosphorylation is a post-translational modification central to signal transduction, that influences cell growth, division, differentiation, cancer development and progression [11, 12]. Protein phosphosites can trigger protein activation or inactivation, and profiling the phosphorylation patterns of proteins can be a powerful tool for understanding key roles in tumor progression and/or drug resistance [13]. Technological advances in the last decade have led to the development of several high-throughput strategies to map the cellular phosphoproteome [14]. Several recent studies examined the phosphoproteome of PC, thereby informing about the activity status of signaling pathways involved in CRPC progression [15–17]. In particular, a recent study integrating phosphoproteomics with transcriptomics and genomics data revealed the diversity of activated signaling pathways in metastatic PC patients, in relation to their resistance to the anti-androgen therapy [18]. This work further demonstrated the utility of combining -omics approaches to better understand PC and CRPC progression.

Here, we used a SILAC-based Mass Spectrometry approach, and identified and quantified the proteomes and phosphoproteomes of four widely used prostate cell lines representative of different cancerous and hormonal status. We first identified a common set of housekeeping proteins highly expressed in all cell lines, and enriched in biological processes related to RNA metabolism and oxidative stress. We further detected that each cell line possesses specific

protein, phosphosite and functional features, in particular related to cellular metabolism, transport and protein localization. In addition, comparing the sensitive and resistant cell lines, we were able to pinpoint potential biomarkers differentially expressed or phosphorylated in the resistant context. Finally, pathway and network-level interpretation of the biomarkers reveal cellular processes associated with resistance, including, among others, an upregulation of cell migration, extracellular processes and epithelial-mesenchymal transition, and a downregulation of the cellular respiration.

## Materials and methods

### Cell culture and SILAC labeling

We cultivated three replicates of four cell lines derived from prostate tissue: PNT1A (ECACC, European Collection of Cell Cultures, England), LNCaP, DU145 and PC3 cell lines (ATCC, American Type Culture Collection (Rockville, MD, USA)). All cell lines were routinely cultured at 37°C in a humidified 5% $CO_2$-95% air atmosphere. They were maintained in Dulbecco's Modified Eagle's Medium (PC3) and RPMI-1640 (Roswell Park Memorial Institute) (Invitrogen, Cergy Pontoise, France), supplemented with 10% fetal bovine serum. Stable Isotope Labelling with Amino acids in Culture (SILAC) labeling of cell lines was carried out according to [19, 20] using SILAC media with 10% dialyzed fetal bovine serum supplemented with $^{13}C_6$$^{15}N_2$-L-lysine (K8) and $^{13}C_6$$^{15}N_4$-L-arginine (R10). Before creating the reference proteome, the incorporation rate of the heavy amino acid was checked for each cell lines using LC-MS/MS and cell extracts were used if this rate reached 95%. Additionally, the interconversion of arginine to proline was checked and found to be negligible. Cells were washed on ice with PBS and collected in a lysis buffer containing 4% SDS, 100 mM Tris-HCl pH7.4, 1 mM DTT (with protease and phosphatase inhibitors cocktails, EDTA-free, ROCHE, usually 1 tablet of each per 10 ml of lysis buffer). Each pellet was resuspended in the lysis buffer and heated to 95°C for 5 min. Viscous lysates were first homogenized mechanically with a syringe and DNAse was added at a 1:40 dilution (benzonase endonuclease, Sigma). Samples were left on ice for 40 min, then centrifuged at 16000 rcf (g) for 25 min. Supernatants were collected in clean Lo-Bind Eppendorf tubes and protein quantitation was done using BCA assay. After cell lysis, the protein extracts from the four heavy cell lines were mixed in equimolar amounts (1:1:1:1), to generate the super SILAC reference proteome which was then aliquoted and stored at -80°C. For proteomics and phosphoproteomics profiling the reference proteome was mixed in equimolar amounts with protein extracts from each non-labeled cells (Fig 1C).

### Proteomes preparation

40 μg of protein extract were loaded on NuPAGE 4–12% bis–Tris acrylamide gels (Life Technologies) to separate proteins, and were stained with Imperial Blue (Pierce, Rockford, IL). Each lane of the gel was cut into 20 bands that were placed in individual Eppendorf tubes. Gel pieces were submitted to an in-gel trypsin digestion using a slightly modified version of the method described by [21]. Briefly, gel pieces were washed and destained using few steps of 100mM ammonium bicarbonate. Destained gel pieces were shrunk with 100 mM ammonium bicarbonate in 50% acetonitrile and dried at room temperature (RT). Protein spots were then rehydrated using 10mM DTT in 25 mM ammonium bicarbonate pH 8.0 for 45 min at 56°C. This solution was replaced by 55 mM iodoacetamide in 25 mM ammonium bicarbonate pH 8.0 and the gel pieces were incubated for 30 min at RT in the dark. They were then washed twice in 25 mM ammonium bicarbonate and finally shrunk by incubation for 5 min with 25 mM ammonium bicarbonate in 50% acetonitrile. The resulting alkylated gel pieces were dried at RT. The dried gel pieces were re-swollen by incubation in 25 mM ammonium bicarbonate

a

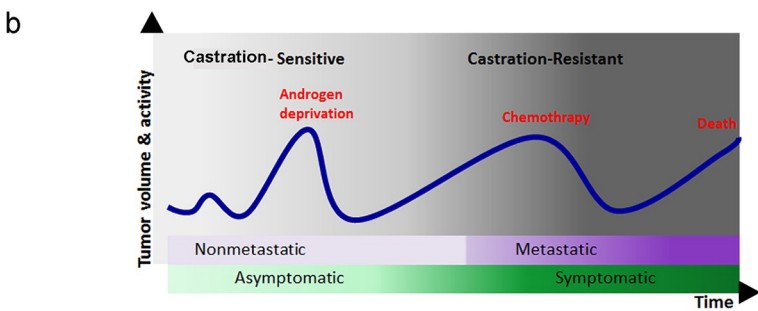

| Cell Line Name | PNT1A | LNCaP | DU145 | PC3 |
|---|---|---|---|---|
| Characterization | SV40 immortalized **normal** prostate | Castration-sensitive prostate cancer | Castration-resistant prostate cancer | Castration-resistant prostate cancer |
| Origin | Prostate epithelium | Lymph node metastasis | Brain metastasis | Bone metastasis |
| Androgen Receptor status | AR- | AR+ | AR- | AR- |

b

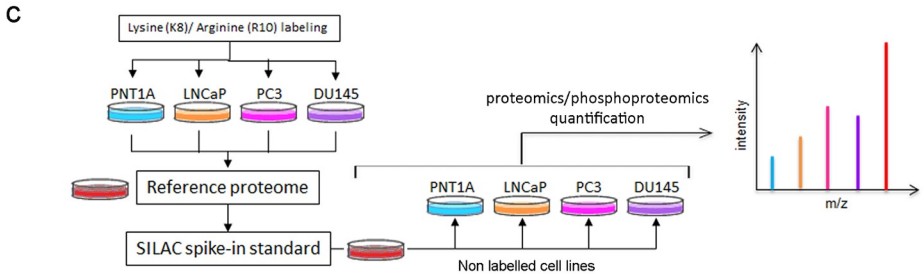

c

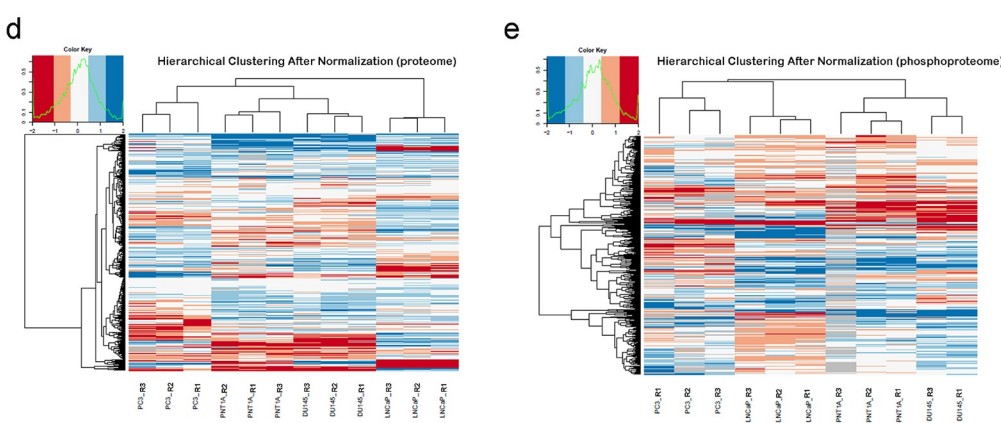

d e

**Fig 1. Overall overview of our study.** (A) Prostate cell lines used in the present study. AR: Androgen Receptor. (B) Prostate cancer progression over time, from localized asymptomatic castration-sensitive to metastatic castration-resistant disease. (C) SILAC Cell line culture preparation, Spike-in and Mass Spectrometry analysis of the proteomes and phosphoproteomes. Figure adapted from [19] (D) Hierarchical clustering of the proteomes and (E) phosphoproteomes normalized expression data in the four cell lines.

pH 8.0 supplemented with 12.5 ng/ml trypsin (Promega) for 1h at 4˚C and then incubated overnight at 37˚C. Peptides were harvested by collecting the initial digestion solution and carrying out two extractions; first in 5% formic acid and then in 5% formic acid in 60% acetonitrile. Pooled extracts were dried down in a centrifugal vacuum system.

## Phosphoproteomes preparation

For each condition, 400 μg of cell lysate implemented with 400 μg of the reference proteome was precipitated using Acetone/Ethanol (sample/Acetone/EtOH 1/4/4 v/v/v) overnight at -20˚C. The acetone-precipitated lysate was resolubilized in 50 mM ammonium bicarbonate, pH 8.0. The soluble proteins were reduced for 45 min at 56˚C with 10 mM dithiothreitol (DTT), and then alkylated for 30 min at RT in the dark with 10mg/ml Iodoacetamide. The protein mixture was then digested with trypsin (1:50 w/w) overnight. Trypsin was quenched by acidification of the reaction mixture with TFA. The peptide mixture was desalted and concentrated on a C18-SepPak cartridge (Waters, Milford, MA) and eluted with 1x 2 mL of 75% acetonitrile (ACN) in 0.1% TFA and dried down. The phosphopeptide enrichment was performed with TiO$_2$ beads 10 μm (Titansphere TIO, GL Sciences, Japan). Titania beads (6 mg) were prepacked in 200 μL pipet tips filled at the orifice with a C8 Empore disk (3M Empore). Prior to loading samples, the titania tips were rinsed with 200 μL of buffer A (3% TFA/70% CH$_3$ CN). Digest samples were reconstituted with 200 μL of loading buffer (buffer A + 1M Glycolic acid). After centrifugation the supernatant was slowly loaded three times onto the titania tip using centrifugation at 300 g for 10 min. The titania beads were sequentially washed with 200 μL loading buffer, twice with 200 μL of buffer A and 200 μL of 0.1% TFA. Bound peptides were eluted with 140 μL of 1% NH$_4$ OH pH 10.5 and dried down with a vacuum concentrator.

## Mass Spectrometry analysis

Samples were reconstituted in 0.1% TFA 4% acetonitrile and analyzed by liquid chromatography (LC)–tandem Mass Spectrometry (MS/MS) using Q-Exactive Mass Spectrometer (Thermo Electron, Bremen, Germany) for proteome and phosphopeptide experiments. For the phosphopeptide experiments, an LTQ-Orbitrap Velos Mass Spectrometer (Thermo Electron, Bremen, Germany) was also used. Mass Spectrometers were on line with a nanoLC Ultimate 3000 chromatography system (Dionex, Sunnyvale, CA). Peptides were separated on a Dionex Acclaim PepMap RSLC C18 column at 37˚C. First, peptides were concentrated and purified on a pre-column from Dionex (C18 PepMap100, 2 cm x 100 μm I.D, 100 Å pore size, 5 μm particle size) in solution A (0.05% trifluoroacetic acid—2% acetonitrile). In the second step, peptides were separated on a reverse phase column from Dionex (C18 PepMap100, 15 cm x 75 μm I.D, 100 Å pore size, 2 μm particle size) at 300 nL/min flow rate. After column equilibration by 4% of solution B (20% water–80% acetonitrile–0.1% formic acid), peptides were eluted from the analytical column by a two steps linear gradient. For proteome analyses, these two steps were 4-25% acetonitrile/H$_2$O; 0.1% formic acid for 40 min and 25-50% acetonitrile/H$_2$O; 0.1% formic acid for 10 min. For phosphopeptide analyses, these two steps were 4-20% acetonitrile/H$_2$O; 0.1% formic acid for 90 min and 20-45% acetonitrile/H$_2$O; 0.1% formic acid for 30 min. For peptides ionisation in the nanospray source, spray voltage was set at 1.5 kV and the capillary temperature at 275˚C. Instrument method for the Q-Exactive was set up in data dependant mode to switch consistently between MS and MS/MS. MS spectra were acquired with the Orbitrap in the range of m/z 300-1700 at a FWHM resolution of 70000 (AGC target at 1e6, maximum IT 120 ms and 250 ms for proteomes and phosphopeptides respectively). For internal mass calibration the 445.120025 ions was used as lock mass. The 12 most intense ions per survey scan (Intensity threshold 1e5) were selected for HCD

fragmentation (AGC target 5e5, NCE 25%, maximum IT 60 ms) and resulting fragments were analysed at a resolution of 17500 in the Orbitrap. Charge state screening was enabled to exclude precursors with unassigned, 1 and > 8 charge states. Fragmented precursor ions were dynamically excluded for 25 s. For phosphopeptides analysis using the LTQ-Orbitrap Velos, the Mass Spectrometer was set as above except for the following parameters. Survey spectra were acquired with a resolution of 60000 (AGC target at 1e6, maximum IT 100 ms) and the 15 most intense precursors ions per cycle were selected for fragmentation by activation of the neutral loss ions (-48.99, -32.66, and -24.49 Thompson relative to the precursor ions) with collision induced dissociation (AGC target 3000, NCE 35%, maximum IT 200 ms). The Mass Spectrometry proteomics data, including search result, have been deposited to the ProteomeXchange Consortium (www.proteomexchange.org) [22] via the PRIDE partner repository with datasets identifiers PXD004970 and PXD004992.

## Protein identification and quantification

Relative intensity-based SILAC quantification was processed using MaxQuant computational proteomics platform, version 1.3.0.5 [23]. First the acquired raw LC Orbitrap MS data were processed using the integrated Andromeda search engine [24]. Spectra were searched against a SwissProt human database (version 2014.02; 20284 entries). This database was supplemented with a set of 245 frequently observed contaminants. The following parameters were used for searches: (i) trypsin allowing cleavage before proline [25]; two missed cleavages were allowed; (ii) monoisotopic precursor tolerance of 20 ppm in the first search used for recalibration, followed by 6 ppm for the main search and 20 ppm for fragment ions from MS/MS; (iii) cysteine carbamidomethylation (+57.02146 Da) as a fixed modification and methionine oxidation (+15.99491 Da) and N-terminal acetylation (+42.0106 Da) as variable modifications; (iv) a maximum of five modifications per peptide allowed; and (v) minimum peptide length was 7 amino acids. The re-quantify option was enabled to search for missing SILAC partners. The quantification was performed using a minimum ratio count of 2 (unique+razor) and the second peptide option to allow identification of two co-fragmented co-eluting peptides with similar masses. The false discovery rate (FDR) at the peptide level and protein level were set to 1% and determined by searching a reverse database. For protein grouping, all proteins that cannot be distinguished based on their identified peptides were assembled into a single entry according to the MaxQuant rules.

## Phosphopeptide identification and quantification

Peptide identification was done similarly than above using MaxQuant software except that serine, threonine, and tyrosine phosphorylation (+79.96633 Da) were allowed as variable modifications.

## Preliminary treatment of the datasets

Statistical analyses were done with the Perseus program (version 1.3.0.5; freely available at www.maxquant.org) from the MaxQuant environment [26]. The relative intensity-based SILAC ratio, iBAQ normalised intensities and peptide intensities were uploaded from the proteinGroups.txt and Phospho(STY)Sites.txt files for proteome and phosphoproteome studies, respectively. Proteins marked as contaminant, reverse hits, and "only identified by site" were discarded.

One DU145 cell line replicate in the phosphoproteome study was discarded due to high divergence. In all other cases, for each experiment and for each cell line, the measurements of three replicates were considered.

## Data analyses

R statistical programming environment [27] was used for the treatment of the proteomic and phosphoproteomic datasets. Expression ratios towards the internal standard were base-2 logarithmized and normalized using z-scores.

**Clustering.** Unsupervised hierarchical clustering using average method was performed for the proteomic and phosphoproteomic datasets based on Euclidean distances of the expression ratio after normalization.

**Identification of the highly-expressed housekeeping proteome.** The abundance of each protein in each cell line was computed as the sum of the IBAQ values of every replicate. The housekeeping proteome was obtained by selecting the 10% most abundant proteins matching across all cell lines.

**Identification of differentially expressed proteins and phosphosites.** We first applied a 1-way ANOVA over the four different cell lines. Benjamini & Hochberg FDR [28] was used for multiple testing corrections, and the threshold for significance was set to 0.01.

Next, to characterize cell line specific protein/phosphosite expression, a t-test was applied to compare the expression value in the three PC cell lines (LNCaP, DU145 and PC3) to the reference non-tumorigenic PNT1A cell line. Benjamini & Hochberg FDR [28] was used for multiple testing corrections, and the threshold of significance set to 0.1.

Pairwise comparisons of protein/phosphosite expression values between the CS (LNCaP) and the CR cell lines (DU145 and PC3) were performed with a t-test, and the threshold of significance set to 0.1 after FDR multiple testing corrections. The results of the pairwise comparisons with the two CR cell lines were combined to define proteins/phosphosites always significantly up- or downregulated in CS as compared to CR.

It is to note that these analyses are conducted with a very stringent filter that select only proteins and phosphosites with at least two over three valid quantification values in all four cell lines. In this context, the proteins identified only in the CR resistant or only in the CS sensitive contexts were discarded, whereas they could be considered as pertinent biomarkers. We then also rescued these potential biomarkers as "CR_only" proteins and phosphosites, having at least two valid expression values in CR and strictly none in CS cell lines and "CS_only" proteins and phosphosites, having at least two valid values in the two CS cell line and strictly none in the CR cell lines.

## Pathway and biological process analyses

**Functional enrichments.** Enrichment Analyses were conducted with G:Profiler [29], and the significance threshold set to 0.01 after FDR multiple testing corrections. The list of 1229 proteins used for quantification analyses was used as statistical background. Additionally, the strong filter option was selected on G:Profiler to display solely the most significant ontology in each ontological group, and reduce annotation redundancy.

**ROMA.** ROMA (Representation and Quantification of Module Activity) is a software focused on the quantification and representation of biological module activity using expression data [30]. The reference gene sets used for this analysis were selected from pathway databases including Reactome [31] and HALLMARK [32]. For each of these pathways, a score corresponding to the weighted sum of the protein expression was computed. The weights are based on the first principal component (PC1). ROMA quantifies the statistical significance of the amount of variance explained by the PC1, and is referred to as the gene set overdispersion. Overdispersed pathways are selected based on a p-value set to 0.01, and the resulting list of pathways can be interpreted as the pathways that contribute significantly to the total expression variance. A detailed presentation of the computational method and use of software can be

found at [30]. For this study, we applied on the proteomic dataset an R implementation of ROMA (https://github.com/Albluca/rRoma), which is an improved version of the initial software. The results are presented as a heatmap where the mean value of the scores was computed by types of cancer cell lines: CS for castration-sensitive and CR for castration-resistant, and scaled between -1 and 1.

**Ingenuity Pathway Analysis (IPA).** Proteomic datasets were also analyzed with Ingenuity Pathway Analysis (IPA) software (Qiagen, http://www.ingenuity.com/) to predict pathway activation or inhibition. The IPA knowledgebase, derived from literature, compute a score based on one-tailed Fisher test. The final score corresponds to the negative log of p-value, and thresholds were set to 0.01.

**Kinase-Substrate Enrichment Analysis (KSEA).** In order to use the KSEA App (https://casecpb.shinyapps.io/ksea/) [33] on the phosphoproteomic datasets, we computed the foldchanges (FC) between DU145 and LNCaP, and between PC3 and LNCaP, using the mean raw expression values of the replicates. We selected the sites where the expression values are over or under-expressed in both CR cell lines in comparison with LNCaP. Finally, we computed the mean of the FC for the 337 Sites, and normalized it between 0 and 1.

We used this list of sites as input for the KSEA App. The kinases with at least 3 targeted phosphosite substrates, and a p-value smaller than 0.05 were considered as significant.

**Network analyses.** We constructed a network encompassing molecular complex interaction data by merging Corum complexes [34] and Hu.MAP complexes [35]. This network contains 8653 nodes and 91500 edges. Then, we fetched interactions between:

- Proteins significantly differentially expressed in CR versus CS;

- Proteins containing phosphosites significantly differentially expressed in CR versus CS;

- Proteins and proteins containing phosphosites identified only in CR or CS contexts (CR_only and CS_only).

The interaction network was represented with Cytoscape [36]. For visualization purposes, the expression values mapped on that network correspond to the mean of the expression of PC3 and DU145 cell lines.

## Results

### Proteomic and phosphoproteomic profiles of prostate cell lines

In order to elucidate PC progression and androgen escape pathway with proteomics and phosphoproteomics identification and quantification, we selected four widely exploited prostate cell lines, namely PNT1A, LNCaP, DU145 and PC3 for proteomic and phosphoproteomic profiling (Fig 1A). These cell lines are representative of normal, cancerous and castration-resistant progression of PC (Fig 1B). The PNT1A benign prostate cell line was established by immortalizing non-tumorigenic human prostate benign epithelial cells by transfection with the SV40 large-T antigen gene [37]. The CS LNCaP cell line was established from metastatic deposit in a lymph node and demonstrates androgen sensitivity [38]. Finally, the two CR tumor cell lines, DU145 and PC3, were established from metastatic deposits (central nervous system and bone/lumbar spine, respectively), lack the AR and are androgen-independent. Moreover, PC3 cells are more tumorigenic and have a higher metastatic potential than DU145 [39]. It is to note that benign PNT1A cell line also lacks the AR. The loss of AR and prostate-associated markers (PSA and PAP) appears to be a consistent feature of immortalized cells of prostatic origin, observed in SV40 immortalized cell lines such as PNT1A [40].

We used SILAC and Mass Spectrometry to identify and quantify the proteomes of these four cell lines [41, 42] (Fig 1C, Materials and methods). We elected the spike-in super SILAC method described by [19, 20]. In this protocol, the protein expression in each cell line is compared to the same reference proteome, thereby maximizing the number of detected proteins. We identified 3219 proteins (S1 Table). We plotted the median iBAQ values considering all the cell lines to estimate the absolute abundance of the 3219 identified proteins, and obtained the expected S-shaped distribution covering six orders of dynamic range of MS signals (Materials and methods and S1 Fig). The most highly expressed proteins include the core histones, tubulins as well as heat shock proteins. Both the most abundant proteins detected as well as the lowest ones have been previously reported in other studies with a similar approach [43]. We kept for further analysis only those proteins containing at least two valid quantification values over the 3 replicates in each cell line. Doing so, we used for subsequent analyses the quantitative expression data of 1229 proteins (S1 Table).

A similar strategy was used to identify and quantify phosphopeptides (Materials and methods). We identified 3746 phosphosites, of which 563 were kept for expression analysis considering the strong filters we defined (S2 Table). These 563 phosphosites correspond to 381 proteins. Overall, 135 proteins were associated with quantitative expression values both at the proteomic and phosphoproteomic levels, with a correlation ranging from 0.43 to 0.62 in each of the four cell lines (S2 Fig). Therefore, the level of phosphorylation of a protein is not strictly correlated to its level of expression, but might also reflect its activity status.

The unsupervised clustering of the quantified proteins and phosphosites first confirms that the cell line replicates cluster together (Fig 1D and 1E). In addition, we observed that the benign PNT1A cell line clusters with the resistant DU145. The genetic instability associated with continuous propagation in culture is a particular problem with benign immortalized cell lines such as PNT1A, in which the insertion of viral DNA drives the cell to replicate continuously [44]. This might explain why its global expression patterns may be similar to that of more malignant cell lines.

## The highly-expressed housekeeping proteome

A large number of proteins are essential in all the cells, suggesting that their expression is crucial for the maintenance of basic functionality and survival [45]. These proteins are often called housekeeping. We focused here on the top 10% most expressed proteins in each cell line, corresponding to 321 proteins. Among those 321 highly expressed proteins, 257 are common to the four cell lines (Materials and methods and S3 Table). This means that 80% of the most expressed proteins are the same in all the four cell lines studied here, and can thereby be defined as the highly-expressed housekeeping proteome.

This housekeeping proteome is enriched in functions related to RNA metabolism and response to oxidative stress (Materials and methods and S5 Table). It contains for instance many RNA binding proteins (mainly from the RPS family) and structural constituents of the ribosome. Eight members of the eukaryotic chaperonin TriC/CCT complex are also highly abundant in all the four cell lines studied.

## LNCaP, DU145 and PC3 cancer cell lines characterization

In a second step, we focused on the differences between the cell lines. We first conducted an ANOVA analysis to identify the proteins and phosphosites with the most variation among the four cell lines (Materials and methods). 46 proteins and 13 phosphosites (corresponding to 13 proteins) are varying significantly among the four cell lines (S3 and S4 Tables). Almost half of the 46 ANOVA-significant proteins play a role in stress response (e.g., DNAJB1, VDAC1,

ZYX, TCEB1), several are involved in actin cytoskeleton organization (e.g., ACTN1, RHOA, PLS3), and 15 proteins are associated with RNA binding (e.g., CCT6A, NOP2, OCT3, HNRNPA2B1). Among the 13 proteins with phosphosites associated with ANOVA-significant variations in the four cell lines, five are cell-adhesion molecule binding (SEPT9, AHNAK, TNKS1BP1, SCRIB, TAGLN2). Of note, Septin-9 (SEPT9), a filament-forming cytoskeletal GTPase, presents significant variations across the cell lines both at the protein and Serine-30 phosphosite levels (S3 Fig). SEPT9 has been shown to be highly expressed in PC and positively correlates with malignant progression [46].

Interestingly, two highly expressed housekeeping proteins are associated with phosphosites differentially expressed between the four cell lines according to the ANOVA analysis. First, TAGLN2 presents a significant variation in the Serine-163 expression (S4 Fig). In liver cancer, this protein has been reported as a putative tumor suppressor and the involvement of its phosphorylation in actin binding and cell migration has been demonstrated [47]. Second, HNRNPA1, involved in the packaging of pre-mRNA, is highly expressed in the four cell lines, but also shows significant differential phosphorylation levels in the Serine-6 (S5 Fig). To our knowledge, a role for HNRNPA1 phosphorylation in PC has not been described previously.

In order to provide insights into the cellular mechanisms that are involved in cell malignant transformation, we then compared protein and phosphosite levels in each of the three cancer cell lines (LNCaP, DU145 and PC3) to the benign PNT1A cell line (Materials and methods). On a global scale, LNCaP clusters apart and appears to be the most divergent cell line (Fig 1D). LNCaP cells display 226 up- and 219 downregulated proteins as compared to PNT1A (S3 Table). Functional enrichment analyses reveal that the proteins upregulated in LNCaP are related to cellular metabolism (Fig 2A and S5 Table). The association of tumorigenesis and metabolism is well established; it is not surprising that a cancer cell, in order to meet its increased requirements of proliferation, displays fundamental changes in pathways of energy metabolism and nutrient uptake [48]. In contrast, the proteins downregulated in LNCaP as compared to PNT1A are enriched in cell recognition and protein/RNA localization processes. Protein and RNA localization mechanisms have shown to play pivotal roles for the presence of specific protein components in cancer cell protrusions, involved in cell migration and invasion [49]. Cell recognition is one of the ways that cells communicate with each other and their environment (adhesion proteins, surface molecules); loss of cell recognition has been shown to lead to cancer development [50]. IPA analysis (Materials and methods) confirmed a high metabolic activity in LNCaP, in particular an upregulation of TCA cycle for aerobic respiration. It further delineates a downregulation in the RAN signaling pathway, central to the nucleo-cytoplasmic transport, with seven downregulated proteins, including RAN and its regulator RANBP1, four importins and one exportin (S5 Table).

The resistant cell line DU145 presents 80 up- and 92 downregulated proteins as compared to PNT1A. Upregulated proteins are enriched in transport and cellular organization processes. Moreover, 61/80 proteins upregulated in DU145 are annotated as extracellular proteins. By contrast, we observed that proteins downregulated in DU145 as compared to PNT1A are enriched in cellular respiration and protein/RNA localization (Fig 2B and S5 Table). IPA analysis confirmed an upregulation of actin and Rho signaling and a downregulation of TCA cycle for aerobic respiration.

Finally, the most tumorigenic cell line, PC3, displays 180 up- and 158 downregulated proteins as compared to PNT1A. The upregulated proteins are enriched in vesicle-mediated transport, as it is the case for the other resistant cell line DU145 (Fig 2C and S5 Table). In recent years, several publications have proposed vesicle-mediated transport as a mechanism to explain the transfer of resistance to drugs among tumorigenic cells [51]. In addition, many proteins upregulated in PC3 are localized in the extracellular exosome. The proteins

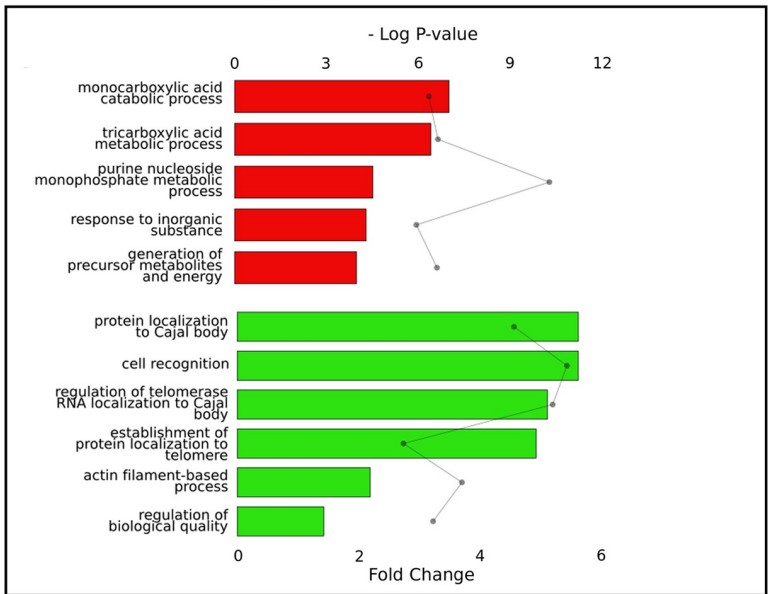

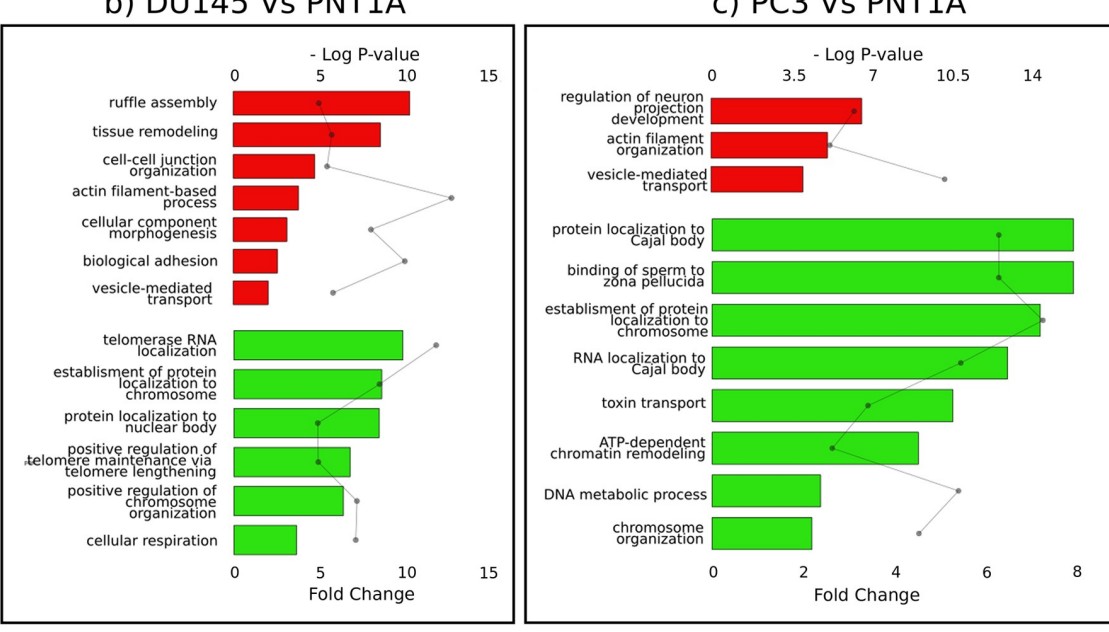

**Fig 2. Functional enrichments of proteins up- and downregulated in PC cell lines.** Bar graphs represent relative fold change of Gene Ontology Biological Processes among (A) LNCaP, (B) DU145, (C) PC3 upregulated proteins (red bars) and downregulated proteins (green bars), as compared to PNT1A cells. Significance is represented in the dot plot by –log (P-values).

downregulated in PC3 are enriched in toxin transport and protein-RNA localization processes. These functional enrichments are complemented by the IPA analysis that revealed strong enrichment in epithelial adherence junction annotation among the upregulated proteins in PC3. Overall, we identified 13 proteins upregulated and 19 proteins downregulated together in LNCaP, DU145 and PC3 cells as compared to PNT1A (S3 Table). We propose that these proteins, differentially expressed in the PC cell lines as compared to the benign cell line, could

constitute markers of oncogenic transformation. The upregulated proteins are almost all annotated for secretion and exosomes (e.g., RAB5B, RAB7A, RPL36A, NES, SRI). It has been recently described that exosomes derived from PC cells modulate the prostatic tumor adjacent environment by inducing, among others, tumor-associated target cells growth [52]. Among the 19 downregulated proteins, several are annotated for regulation of protein stability and chaperone-mediated protein folding, and almost half are involved in DNA metabolism. Overall, many proteins of the chaperonin TriC/CCT folding complex, which were observed as highly abundant in all cell lines and thereby classified as housekeeping, are also underexpressed in the three cancer cell lines as compared to PNT1A. The TriC/CCT chaperonin complex directly modulates the folding and activity of as many as 10% of cytosolic client proteins [53, 54]. Recently, the TRiC/CCT complex was also shown to be required for maintaining the wild-type conformation of the tumor suppressor p53 [55]. The downregulation of this chaperone complex could promote the oncogenic functions of p53, such as cancer cell migration and invasion.

We reproduced the cell line characterization protocol for phosphosites, thereby identifying 146 up- and 98 downregulated phosphosites in LNCaP, 5 up- and 3 down in DU145, and 82 up- and 44 down in PC3, as compared to PNT1A. No functional enrichments were significant for the corresponding proteins. Nevertheless, two proteins are associated with phosphosites significantly deregulated in all three PC cell lines as compared to PNT1A. First, TP53BP1 (tumor protein p53 binding protein 1) phosphosites Serine-500 and Threonine-1056 are downregulated in LNCaP. TP53BP1 Serine-500 phosphosite is also downregulated in DU145, and the Threonine-1056 phosphosite downregulated in PC3, as compared to PNT1A. This TP53BP1 protein is well known to be involved in DNA Damage Response (DDR) and its phosphorylation could be a marker of malignant transformation [56]. Previously published studies described TP53BP1 phosphorylation necessary for recruitment to DNA double strand breaks [57]. In this context, a downregulation of TP53BP1 phosphorylation, as we observed in the three PC cell lines, could lead to impaired DDR. Second, the DEAD-box RNA helicase 10 (DDX10) Serine-539 phosphosite is significantly upregulated in LNCaP, DU145 and PC3 as compared to PNT1A. DDX10 is an ATP-dependent RNA helicase [58], but, to our knowledge, little is known about its phosphorylation and function in cancer. Other members of the same family of RNA helicases have been well described, and the phosphorylation of DDX p68 is reported to be associated with cancer development and cell proliferation [59]. Interestingly, the phosphosite Serine-539 that we identified as upregulated in PC cell lines is one of the known post-translational DDX modification sites [60]. Thus, our approach allowed us identifying a well-known cancer-related phosphosite, as well as another potential new candidate.

### Identification of resistance markers

One of the features of PC is, in most cases, its progression to highly aggressive and incurable CR disease after androgen deprivation therapy. Identifying resistance biomarkers is essential to guide the development of new therapeutic strategies and avoid drug resistance. In order to identify proteins and processes potentially involved in resistance, we compared protein expression levels in CS LNCaP cell line, with CR DU145 and PC3 cell lines. We found 135 proteins upregulated and 135 downregulated in CR as compared to CS cell line, and propose them as resistance biomarkers (S3 Table). Protein biomarkers upregulated in the CR contexts are functionally enriched in processes related to cell-cell adhesion and external communication (Fig 3A and S5 Table). This finding is in accordance with previously published studies demonstrating the involvement of these processes in invasion and metastasis, features for which CR cells have a higher potential [61]. Conversely, proteins downregulated in CR are enriched in cellular

## a) DU145 & PC3 Vs LNCaP

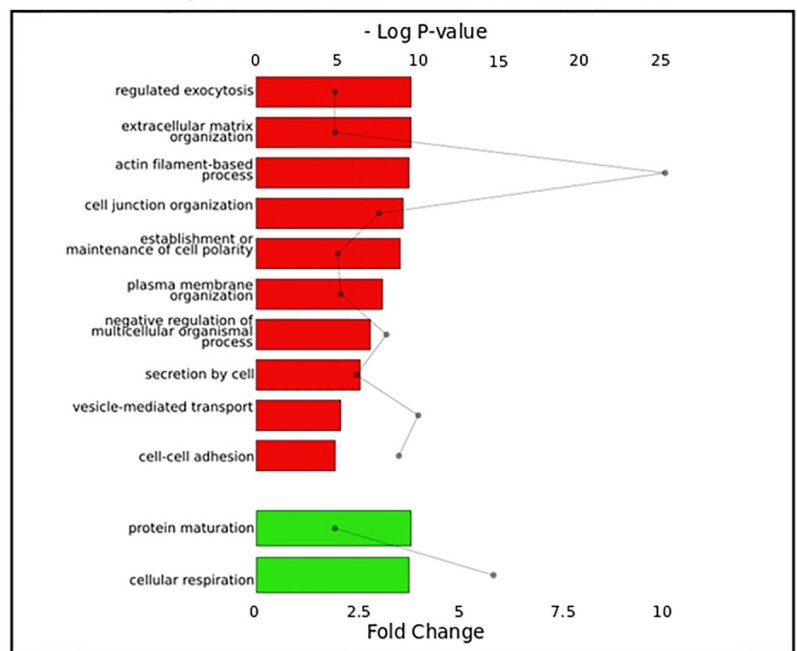

## b) DU145 & PC3 Vs LNCaP

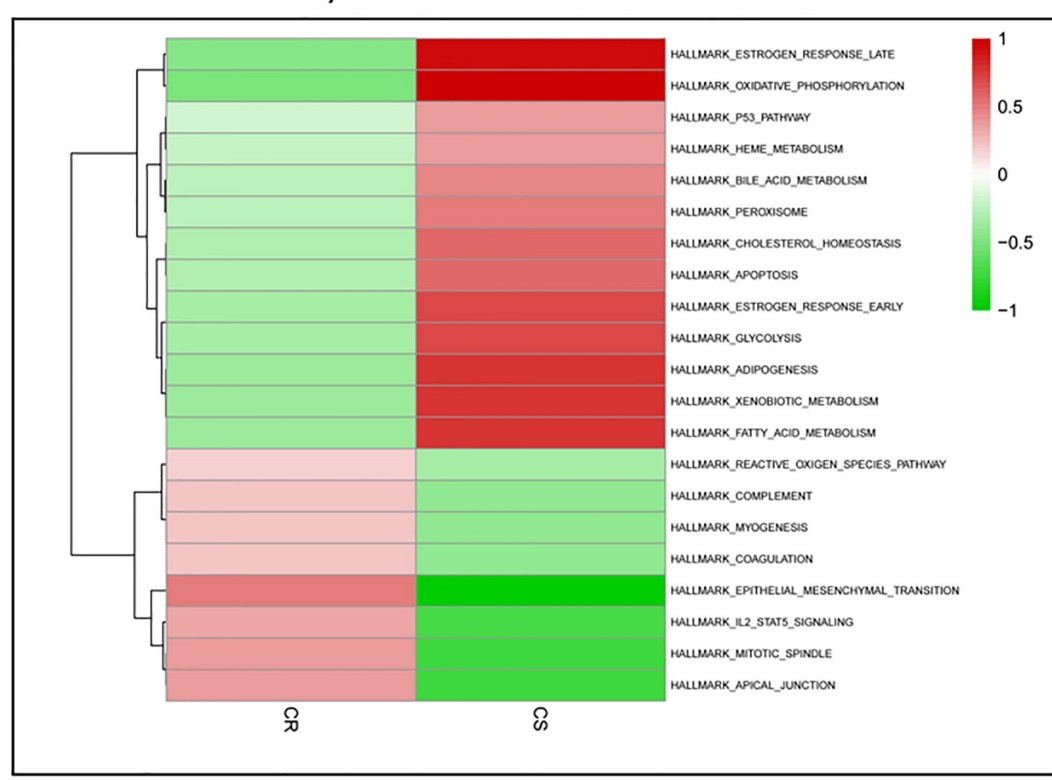

**Fig 3. Functional enrichments of protein resistance biomarkers.** (A) Bar graphs represent relative fold change of Gene Ontology Biological Processes among proteins upregulated (red bars) and downregulated (green bars) in Castration Resistant cell lines DU145 and PC3 as compared to castration-sensitive LNCaP cell line. Significance is represented in the dot plot by –log (P-values). (B) Clustered heatmap of ROMA pathway analysis. The color intensities correspond to the values of the scores of each signaling pathway (red, upregulated; green, downregulated).

respiration and protein maturation processes. The downregulation of cellular respiration in the CR context could highlight the Warburg effect [62], in which castration-resistant progression would be associated with a switch from oxidative respiration to glycolysis as primary energy source. The ROMA pathway analysis tool [30] also points to a downregulation in CR cells of oxidative phosphorylation and metabolic pathways such as fatty acid metabolism, as well as signaling pathways related to p53 and apoptosis (Fig 3B). Conversely, it reveals an upregulation of the epithelial-mesenchymal transition (EMT) and reactive oxygen species (ROS) pathways. EMT refers to the morphological and functional alterations involved in cancer invasion [63]. Finally, IPA analysis points to an upregulation of actin cytoskeleton and Rho signaling in CR cells, and further identifies an upregulation of Integrin Signaling and Calpain protease signaling.

Phosphoproteomics data reveal 41 phosphosites upregulated and 40 downregulated in CR versus CS, which we also predict as resistance biomarkers (S4 Table). The 41 upregulated phosphosites concern essentially nuclear proteins involved in functions such as transcription regulation, genome stability and RNA processing (e.g., SMARCC1, SRRM1, SRRM2, SSB). The deregulation of these processes, and their implication in cancer development and progression, have been largely documented [64]. Moreover, 2 kinases are hyper-phosphorylated in the resistant context. First, the Serine/threonine-protein kinase N2 (PKN2), which plays a role in the regulation of cell cycle progression, actin cytoskeleton assembly, cell migration, cell adhesion, tumor cell invasion and transcription activation signaling processes. It was recently shown to be phosphorylated by the PI-3 Kinase pathway and implicated in PC progression [65]. Second, the nuclear receptor binding protein (NRBP1), which is involved in subcellular ER-Golgi trafficking. To our knowledge, a role of its phosphorylation status in PC has not been described previously.

The 40 downregulated phosphosites concern mainly proteins involved in cell migration and invasion, such as PLEC, AHNAK, ESYT1 and ZYX. A group of kinases sharing the same identified peptide and that consequently cannot be distinguished with the MS experiment (CDK2;CDK3;CDK1;CDC2) shows a decrease in phosphorylation activity in the CR context.

Kinase-Substrate Enrichment Analysis (KSEA [33], Materials and methods) predicted the high activity of 3 kinases, namely CDK1, MAPK13 and MAPK3, with 9, 4 and 3 targeted phosphosites that present significant changes in the CR context, respectively (S4 Table). For instance, the Serine-25 and Serine-38 of the stathmin protein (STMN1) are targets of the three kinases. The STMN1 protein displays a complex pattern of activity and phosphorylation in cancers [66]. The sequestosome 1 protein (SQSTM1) Threonine-269 and Serine-272 are targets of both CDK1 and MAPK13.

Another interesting set of putative biomarkers can be derived from the proteins and phosphosites that have been identified in the MS experiment, but that were not further considered for quantification analyses because of the strong filtering criteria we have defined. We thus rescued the proteins and phosphosites that have been identified in at least 2 replicates in the CS cell line but that are completely absents in the CR cell lines, and vice-versa (Materials and methods). This concerns 140 proteins and 5 phosphosites that are identified only in the CR cell lines, and 8 proteins and 108 phosphosites that are identified only in the CS cell line. Focusing particularly on kinases, 8 of them are identified only in the CR cell lines (CALM1, EGFR, EIF2AK2, EPHA2, HK2, PIK3R4, PPP4C, ROCK2). A majority of these kinases are involved in response to stress. Two other kinases are associated with phosphosites identified only in the CR contexts (PRPF4B, TAOK1). TAOK1 is particularly appealing as it activates the Hippo pathway involved in cellular homeostasis [67]. Finally, it is to note that some phosphosites associated to significantly different levels of phosphorylation are found in proteins that are quantified by our approach, and not differentially expressed in the ANOVA. These might

represent functionally relevant candidates. These include 12 proteins (DKC1, BCLAF1, SRRM2, NAP1L4, CLNS1A, TJP1, API5, SSB, SQSTM1, DHCR7, NCBP1).

## Proteome and phosphoproteome integration in a molecular network

We finally sought to provide a larger-scale functional interpretation of resistance-associated candidate biomarkers. The separated analysis of the proteomics and phosphoproteomics datasets provided one-dimensional views of cellular processes. We expect to obtain a comprehensive perspective of cellular processes and their interplays by integrating the information about protein abundances, activation status and molecular interactions [68, 69]. Toward this goal, we devised a network-guided integration of CS and CRPC cell lines proteome and phosphoproteome, by mapping the candidate biomarkers to molecular complex interaction data (Materials and methods). The resulting network is composed of 356 nodes and 1161 edges, including a large connected component encompassing 194 nodes and 1098 edges (Fig 4). The network reveals the links between up- and downregulated proteins, up- and downregulated phosphosites and corresponding proteins, as well as the links between the proteins and phosphosites that were identified by the MS approach only in the CR or CS contexts. At-a-glance, we can observe that the network is organized around several strongly connected subnetworks.

First, we identified a cell migration/invasion subnetwork, which is composed mainly of upregulated proteins in CR cells (e.g., ANXA2, IQGAP1, ACTN4, TWF1, MYO1B, CORO1C, ARPC4) (Fig 4). It contains in particular the plectrin protein (PLEC), overexpressed and hyper-phosphorylated in CR; this protein is known to interlink cytoskeleton elements and promote cancer cell invasion and migration [70]. Indeed, it was shown that along with vimentin intermediate filaments, plectrin provide a scaffold for invadopodia formation, facilitating cancer cell invasion extravasation for metastasis [71]. Recently, [72] demonstrated that

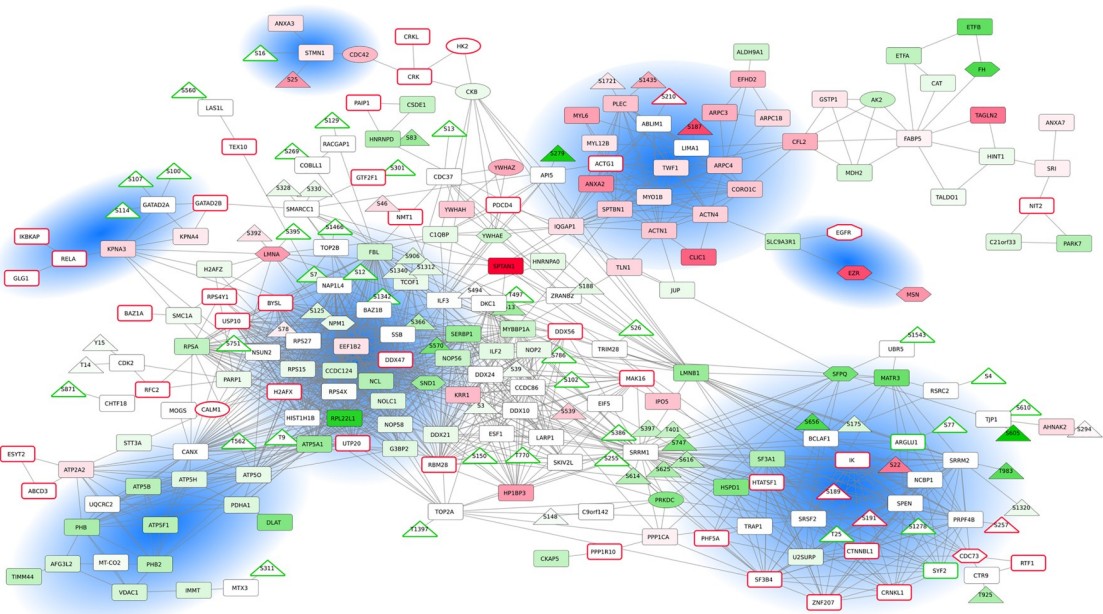

**Fig 4. Network of CR biomarker interactions.** Proteins (boxes) and phosphosites (triangles) significantly upregulated or downregulated in the CR context are mapped in red or green, respectively, with color intensities related to fold-changes. For visualization purposes, the expression values correspond to the mean of the expression of PC3 and DU145 cell lines. Proteins and phosphosites identified only in CR (DU145 and PC3) or CS (LNCaP) cells lines are squared in red and green, respectively.

upregulation of vimentin and plectrin expressions positively correlates with the invasion and metastasis of androgen-independent PC cells. Another interesting member of this complex is ACTG1 (actin gamma-1), which is not identified, and thereby might be not expressed, in CS cells. ACTG1 is involved in cell motility/cytoskeleton maintenance and cancer cell migration. ACTG1 was shown to induce cancer cell migration in lung cancer cells and hepatocellular carcinoma cells [73]. To date, there is no report concerning ACTG1 involvement in PC. The subnetwork also contains components of the Arp2/3 complex (ARPC1B, ARPC3, ARPC4) involved in the regulation of actin polymerization.

A smaller subnetwork, composed of interactions between EZR, MSN, SLC9A3R1 and EGFR, is located close to the larger migration subnetwork. Ezrin (EZR) and moesin (MSN) are scaffolding proteins that are involved in crosslinking cytoskeletal and membrane proteins. Ezrin is involved in oncogenesis through these interactions [74], and it was also shown recently that Ezrin can increase the oncogenic functions of EGFR [75]. SLC9A3R1 is also a scaffold protein that connects plasma membrane proteins with members of the ezrin/moesin/radixin family linking them to the actin cytoskeleton and regulating their surface expression [76].

We also identified a small subnetwork of interacting proteins involved in actin cytoskeleton regulation (e.g., STMN1, CDC42, CRLK1). Intriguingly, we found that stathmin1 (STMN1) was both hyper- and hypo-phosphorylated in CR cells. This protein is associated with cancer metastasis and exhibits a complicated phosphorylation pattern in response to various extracellular signals [77].

We next focused on a small subnetwork composed of proteins underexpressed in the CR context. It contains prohibitin (PHB), a putative tumor suppressor protein involved in the inhibition of DNA synthesis and regulating proliferation, and prohibitin-2 (PHB2), a mediator of transcriptional repression by nuclear receptors, also potentially involved in mitochondrial respiration. Indeed, the subnetwork also contains the VDAC1 mitochondrial membrane and plasma membrane channel, involved in apoptosis. The role of this subnetwork is unclear, but the proteins are depicted as members of the same complexes in the Hu.map dataset [35]. The subnetwork is tightly linked to another subnetwork containing many mitochondrial membrane ATP synthase proteins (e.g., ATP5F1, ATP5B, ATP5H), also downregulated in CR cell lines.

A heterogeneous subnetwork is composed of many proteins involved in splicing and RNA processing, that are either up- or downregulated in CR cells (Fig 4). Splicing events control gene expression and their alterations have been shown to play a role in cancer [78] and specifically in PC [79]. Fine regulation of expression and/or phosphorylation status determines whether a splicing factor functions as a splicing repressor or activator [80, 81]. The subnetwork contains the hypo-phosphorylated splicing factors SRRM1 (a highly phosphorylated protein under normal conditions [82]) and SRRM2. It also contains NCBP1, which is identified as hyper-phosphorylated, and PRPF4B kinase and SPEN that were both hypo-phosphorylated. The subnetwork also incorporates pre-mRNA splicing factor SYF2, absent in CR cells, and SF3A1, TRAP1 and HSPD1 proteins that are downregulated in CR cells. The protein phosphatase 1 (PPP1CA) is contrarily upregulated. Interestingly, we can also observe many proteins identified in the CR cell lines and absent in the CS cell line, all involved in RNA processing and splicing (IK, ZNF207, CTNNBL1, CRNKL1, SF3B4, HTATSF1, PHF5A, PPP1R10). PPP1R10, the Ser/Thr-protein phosphatase-1 regulatory subunit 10 is only expressed in CR cells and is absent in CS cells. It has been shown that certain Ser/Thr-specific protein phosphatases are required for catalytic steps of pre-mRNA splicing [83].

We then emphasize a large and highly connected component (Fig 4) composed of proteins implicated in DDR. It contains protein biomarkers downregulated in CR cells (NPM1,

NOLC1, RPL22L1, FBL, G3BP2), but also several proteins identified only in the CR cell lines, namely H2AFX, kinase CALM1, DDX47, UTP20, USP10, BYSL. All these proteins interact with single-strand DNA-binding protein and are involved in DNA repair and genome stability [84]. DNA repair and DDR are known to be defective in PC and lead to genome instability [85]. Interestingly, several of the proteins of this subnetwork (e.g., UTP20, BYSL, RPL22L1, NOLC1) are known for their role in RNA processing. There is an increasing number of studies demonstrating the involvement of RNA processing factors in DDR [86, 87]. For instance, NOLC1 (nuclear and coiled-body phosphoprotein-1) is a regulator of RNA polymerase I and has been recently shown to regulate the nucleolar retention of TERF2, inducing telomeric DNA damage [88].

A closer look into this molecular network allowed us to pinpoint several interesting smaller subnetworks. For instance, we noticed a small subnetwork composed of interacting proteins RELA, IKBKAP, GLG1, KPNA3, KPNA4. Importin subunits alpha-4 (KPNA3) and alpha-3 (KPNA4) are involved in nuclear transport of NF kappa B [89], and an elevated activity of the NF-kappa B signaling in CRPC is positively correlated with poor prognosis in CRPC [90]. Close to this subnetwork, GATAD2B is known to form a homodimer with GATAD2A and the complex is part of a highly conserved chromatin-remodeling complex, the NuRD complex associated with DNA damage-induced transcription repression but also metastasis and EMT [91, 92]. This subnetwork is also linked to the SWI/SNF complex subunit SMARCC1, which contains downregulated phosphosites in PC3 and DU145 cells. SMARCC1 positively regulates transcription and was previously shown to induce PC survival [93]. It interacts with proteins associated with phosphosites only detected in CS cells (transcriptional elongation factor TRIM28, transcription kinase BAZ1B and TOP2B), as well as with proteins only identified in PC3 and DU145 cells (e.g., GATAD2B, GTF2F1), all involved directly or indirectly in transcription regulation. The transcriptional reprogramming in PC progression has been extensively studied, as it is one of the hallmarks of CRPC [3, 94–96].

## Discussion

We here generated and explored a SILAC proteomics and phosphoproteomics dataset of prostate cell lines. We selected the PNT1A, LNCaP, DU145 and PC3 cell lines first because they are frequently used in PC research [97, 98], and molecular profiling and comparisons would thereby be highly valuable for researchers using them in routine. In addition, the elected cell lines are representative of normal, cancerous and CR prostate tissue, and therefore reflect progression of the disease.

We decided to monitor proteome and phosphoproteome jointly as they can give a complementary picture of the molecular dynamics of the cells. Phosphoproteome characterization provides insights into proteins' phosphorylation levels, which are not strictly correlated to proteins level of expression, but also reflect protein activity status [99]. Molecular characterization at these two levels of information is a clear advantage in our study. We identified 3 219 proteins at 1% FDR, and after several filtering steps, we performed subsequent functional explorations on 1229 proteins. We elected this conservative approach in order to avoid imputation of missing values and ensure the results of the statistical analyses. On the phosphoproteomics side, we identified 3746 phosphosites among which 563 are used for subsequent analyses. We explored the proteins and phosphosites associated to the four PC cell lines; these profiles and associated cellular processes could be related to underlying biology of the cell lines (Fig 1, S1 Fig, S1 and S3 Tables). We further compared protein and phosphosite levels in each of the three cancer cell lines (LNCaP, DU145 and PC3) to the benign PNT1A cell line (Fig 2, S3 and S5 Tables). Doing so, we proposed several candidates that could constitute markers of

oncogenic transformation. Notably, two proteins are associated with phosphosites significantly deregulated in all three PC cell lines as compared to PNT1A: TP53BP1, well-known cancer-related phosphosites downregulated in all three PC cell lines and DDX10, a potential new candidate, upregulated in PCa cell lines. We also highlighted differentially expressed proteins and processes potentially involved in resistance, by comparing the sensitive LNCaP cell line to the two resistant DU145 and PC3 cell lines (Fig 3). Interestingly, we identified 12 proteins associated to significant differential phosphorylation but not different protein levels, including several RNA binding proteins, sequestosome-1 protein (involved in autophagy in relation to many crucial signalling pathways), as well as DHCR7, an enzyme involved in cholesterol metabolism, and TJP1, a tight junction protein. Finally, we proposed an integrated mapping of protein abundances, activation status and molecular interactions, towards functional interpretation of resistance-associated candidate biomarkers (Fig 4). We observed that the network is organized around several strongly connected subnetworks (cell migration/invasion subnetwork, actin cytoskeleton regulation, DNA synthesis and regulating proliferation, splicing and RNA processing, DDR) and several interesting smaller subnetworks (transcription regulation, nuclear transport). Several of these proteins and phosphosites are already related to cancer resistance in general, and some specifically to PCa. Overall, this analysis represents a valuable resource that could be used as a starting point for further hypothesis and experimental investigations.

## Conclusion

The complex nature of PC, due to its clinical and molecular heterogeneities, makes it difficult to determine a perfect model representing tumor development, and precludes easy correlation of carcinoma cell lines with specific stages of PC. Nevertheless, PC cell lines routinely used for the last three decades have provided valuable resources for understanding important functional molecular mechanisms involved in this disease. In the present study, we used four cell lines that constitute a gold standard for pre-clinical studies of PC progression. We conducted a large SILAC-based Mass Spectrometry identification and quantification of peptides and phosphopeptides of prostate benign, castration-sensitive (CS) and castration-resistant (CR) cells, and characterized housekeeping, cell line, cancer and resistance associated proteomes and phosphoproteomes.

## Supporting information

**S1 Fig. Dynamic range of the prostate cancer proteome.** (a) Ranking of the absolute abundance using the IBAQ intensity. The expression values of every protein in the three replicates of the four studied cell lines were considered. (b) Zoom on the left box in (a) displaying the 25 less abundant proteins. (c) Zoom on the right box in (a) displaying the 25 most abundant proteins.
(TIF)

**S2 Fig. Correlation between proteomic and phosphoproteomic expression values.** We computed for each cell line the correlation between the expression values of the 135 proteins that were quantified both at the proteomic and the phosphoproteomic levels. For proteomics data, we computed the mean of the three replicated. For phosphoproteomics data, we computed the mean for all the phosphosites belonging to the same protein.
(TIF)

**S3 Fig. Expression Profiles associated with Septin-9 (SEPT9).** (a) Boxplot showing the SEPT9 protein expression values in the four cell lines under study. (b) Boxplot revealing the

SEPT9 Serine-30 phosphosite expression values in the four cell lines under study.
(TIF)

**S4 Fig. Expression Profiles associated with TAGLN2.** (a) Boxplot showing the TAGLN2 protein expression values in the four cell lines under study. (b) Boxplot revealing the TAGLN2 Serine-163 phosphosite expression values in the four cell lines under study.
(TIFF)

**S5 Fig. Expression Profiles associated with HNRNPA1.** (a) Boxplot showing the HNRNPA1 protein expression values in the four cell lines under study. (b) Boxplot revealing the HNRNPA1 Serine-6 phosphosite expression values in the four cell lines under study.
(TIFF)

**S1 Table. Proteins identified and quantified in the MS experiment.** Sef of proteins identified in the MS experiment, and subset of filtered proteins associated with at least 2 valid quantification values in all four cell lines, which were kept for expression analyses.
(XLSX)

**S2 Table. Phosphosites identified and quantified in the MS experiment.** Set of phosphosites identified in the MS experiment, and subset of filtered phosphosites associated with at least 2 valid quantification values in all four cell lines, which were kept for expression analyses.
(XLSX)

**S3 Table. Subdatasets of interest in proteomic expression analyses.** It contains the ANOVA-significant proteins, the proteins up- and downregulated in the three prostate cancer cell lines as compared to the benign PNT1A cell line, the proteins up- and downregulated in the castration-resistant (CR: DU145 and PC3) cell lines as compared to the castration-sensitive (CS: LNCaP) cell line, and the proteins identified only in the CR or CS contexts (CR_only, CS_only).
(XLSX)

**S4 Table. Subdatasets of interest in phosphoproteomic expression analyses.** It contains the ANOVA-significant phosphosites, the phosphosites up- and downregulated in the three prostate cancer cell lines as compared to the benign PNT1A cell line, the phosphosites up- and downregulated in the castration-resistant (CR: DU145 and PC3) cell lines as compared to the castration-sensitive (CS: LNCaP) cell line, and the phosphosites identified only in the CR or CS contexts (CR_only, CS_only). It further contains the results of the KSEA analysis.
(XLSX)

**S5 Table. Functional enrichment analyses results.** Raw results of the functional enrichment analyses with G:profiler and Ingenuity Pathway Analyses (IPA).
(XLSX)

## Acknowledgments

The authors thank Christine Brun, Andreas Zanzoni and all the partners of the Hsp27BioSys project for fruitful discussion. The differential proteomic analyses were done using the Mass Spectrometry facility of Marseille Proteomics (http://marseille-proteomique.univ-amu.fr/) supported by IBISA (Infrastructures Biologie Santé et Agronomie), the Cancéropôle PACA, the Provence-Alpes-Côte d'Azur Region, the Institut Paoli-Calmettes and the Centre de Recherche en Cancérologie de Marseille.

## Author Contributions

**Conceptualization:** Palma Rocchi, Luc Camoin, Anaïs Baudot.

**Formal analysis:** Maria Katsogiannou, Alberto Valdeolivas, Elisabeth Remy, Laurence Calzone, Luc Camoin, Anaïs Baudot.

**Funding acquisition:** Palma Rocchi, Anaïs Baudot.

**Investigation:** Maria Katsogiannou, Jean-Baptiste Boyer, Alberto Valdeolivas, Elisabeth Remy, Palma Rocchi, Luc Camoin, Anaïs Baudot.

**Methodology:** Maria Katsogiannou, Jean-Baptiste Boyer, Elisabeth Remy, Laurence Calzone, Stéphane Audebert, Luc Camoin, Anaïs Baudot.

**Project administration:** Palma Rocchi, Luc Camoin, Anaïs Baudot.

**Resources:** Stéphane Audebert, Luc Camoin.

**Software:** Alberto Valdeolivas, Laurence Calzone.

**Supervision:** Luc Camoin, Anaïs Baudot.

**Validation:** Anaïs Baudot.

**Visualization:** Maria Katsogiannou, Alberto Valdeolivas, Laurence Calzone, Luc Camoin, Anaïs Baudot.

**Writing – original draft:** Maria Katsogiannou, Alberto Valdeolivas, Laurence Calzone, Luc Camoin, Anaïs Baudot.

**Writing – review & editing:** Maria Katsogiannou, Alberto Valdeolivas, Luc Camoin, Anaïs Baudot.

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
