## [Decision Letter · Decision Letter 0]

13 Aug 2019

PONE-D-19-20799

Integrative proteomic and phosphoproteomic profiling of prostate cell lines

PLOS ONE

Dear Dr Valdeolivas

Thank you for submitting your manuscript to PLOS ONE. After careful consideration, we feel that it has merit but does not fully meet PLOS ONE’s publication criteria as it currently stands. Therefore, we invite you to submit a revised version of the manuscript that addresses the points raised during the review process.

A significant number of concerns have been raised by 3 expert Reviewers; the comments are technical, scientific and do concern also the text.  

We would appreciate receiving your revised manuscript that addresses all comments.

To enhance the reproducibility of your results, we recommend that if applicable you deposit your laboratory protocols in protocols.io, where a protocol can be assigned its own identifier (DOI) such that it can be cited independently in the future. For instructions see: http://journals.plos.org/plosone/s/submission-guidelines#loc-laboratory-protocols

We look forward to receiving your revised manuscript.

Kind regards,

Lucia R. Languino, Ph.D.

Academic Editor

PLOS ONE

"This work was supported by the French "Plan Cancer 2009-2013" (Systems Biology call). The funders had no role in study design, data collection and analysis, decision to publish, or preparation of the manuscript."

We note that one or more of the authors are employed by a commercial company: 'ProGeLife, Marseille, France'.

Reviewers' comments:

Reviewer's Responses to Questions

**Comments to the Author**

1. Is the manuscript technically sound, and do the data support the conclusions?

Reviewer #1: No

Reviewer #2: Yes

Reviewer #3: Partly

2. Has the statistical analysis been performed appropriately and rigorously? 

Reviewer #1: Yes

Reviewer #2: Yes

Reviewer #3: Yes

3. Have the authors made all data underlying the findings in their manuscript fully available?

Reviewer #1: Yes

Reviewer #2: Yes

Reviewer #3: Yes

4. Is the manuscript presented in an intelligible fashion and written in standard English?

Reviewer #1: Yes

Reviewer #2: Yes

Reviewer #3: Yes

5. Review Comments to the Author

Reviewer #1: There are several studies in the literature about prostate cancer proteomics. However, in some cases these previous publications yielded data with limited reproducibility. It is sometimes difficult to distinguish between data which have implications in the clinic and others. Although data quality is accptable, there are some questions about the conclusions.

Specific points:

1. the reason for comparison of sensitive and resistant cell lines are not clear. This could imply that androgen receptor expression decreases during tumor progression what is not the case. It is difficult to draw any conclusion from this comparison.

2. the descriütion of human cell lines in results is not necessary. They are very well known in scientific community.

3. In the Results section, there are too many repetitions of that what has alrready been presented in Materials and Methods.

4. it is not clear whether this manuscript could confirm previous findings in the proteomic field.

Reviewer #2: M Katsogiannou et al presented a study that aimed to investigate and explore the proteome and phosho-proteome of four well established prostate cancer cell lines. Utilizing a SILAC-based Mass Spectrometry approach, the data show set of proteins that are commonly and highly expressed in all four cell lines, as well as differentially expressed proteins between castrate-resistant and castrate-sensitive cells. Other comparisons such as proteins up-regulated in cancer cell lines compared to non-tumorigenic cell is demonstrated. Phospho-proteomic data is also presented for the cell lines including presentation of the proteome and phosphor-proteome in a molecular network to identify candidate biomarkers for Prostate cancer. Overall, the study is well-done and provides a starting point to further explore the role of potential candidate proteins identified through this study in prostate cancer. Although the study adds some new potential proteins to the list as target molecules and biomarkers, it reconfirms a lot of information that is already published in the field.

1)There are other studies that have used proteomic -based approaches on prostate cancer cells or conditioned media from prostate cancer cells? What is the novelty of the approach used in this study?

2) Please show if the data can be used for miRNA target prediction? It would be a good addition to the study as miRNA’s have great potential as biomarkers in cancer.

3) Add PCA plot to show that each cell line has a unique protein and phosphor-protein signature and add Venn-diagrams to show common and unique expressors for a) cancer cell lines compared to non-tumorigenic cells b) castrate resistant cells compared to castrate sensitive cells.

4) Are there common expressors and unique expressors between DU145 and PC3. This could perhaps help identify candidate biomarkers and targets for highly metastatic and aggressive disease.

5) Show box-plots for expression levels of TAGLN2 and HNRNPN1 in an additional figure. Validation of differences in protein levels must be shown.

5) The resolution is images in the figures needs to be increased. Figures are hard to read as of now.

Reviewer #3: The manuscript from Katsogiannou et al. showed a large SILAC-based Mass Spectrometry experiment that allowed to map the proteomes and phosphoproteomes of PNT1A, LNCaP, DU145 and PC3 prostate cancer cell lines, and reveal different signaling networks associated with the cellular context of each cell line, possibly reflecting the pathological features of human Prostate Cancer (hormonal status, ability to metastatize etc.).

The experimental data are strong, rigorous and well presented, in particular the deep comparison of the four cell lines for the identification of the housekeeping proteome vs the most significant variations across the samples.

However, some critical points emerged, and should be clarified/investigated more in depth:

1. The androgen-dependent/castration-sensitive cell line LNCaP has been compared with the androgen-independent/castration-resistant cells DU145 and PC3. However, the best in vitro models for comparing these two PrCa conditions would be LNCaP vs C4-2. Why the authors did exclude C4-2 cells from their high-throughput analysis?

2. The ANOVA analysis of the proteomics/phosphoproteomics data highlighted several proteins/phosphosites that vary significantly in various comparisons (benign vs malignant, CR vs CS etc.). For some of these variations, the authors even claimed that “these proteins could constitute markers of oncogenic transformation”. To support this kind of statements, the authors should provide “wet-lab” validations of their high-throughput results, at least on representative targets among those described in the text (e.g. Septin-9, TAGLN2, HNRNPA1, RAB5B/RAB7A, TriC/CCT complex, TP53BP1 pSer-500 and pThr-1056, DDX10 pSer-539 etc). This type of validation would also help the authors focusing on the most important pathways, rather than leaving the reader with a comprehensive description of all signaling networks potentially involved in the regulation of PrCa malignant progression.

Minor points:

1. Page 8, Lanes 303-305): DU145 are derived from CNS metastasis, and PC3 from bone metastasis. Please revise the sentence.

2. Figure 2: it should be useful to include a “title/legend” to each bar graph (e.g. LNCaP vs PNT1A in the panel A), showing the comparisons as described in the figure legend text.

3. Figure 3: similar to the previous point, it should be useful to include a “visible” title/legend to each panel (e.g. DU145/PC3 vs LNCaP in the panel A, and CR vs CS on top of the panel B).

6. PLOS authors have the option to publish the peer review history of their article (what does this mean?). If published, this will include your full peer review and any attached files.

Reviewer #1: No

Reviewer #2: No

Reviewer #3: Yes: Marco Trerotola, PhD

Laboratory of Cancer Pathology

Center for Advanced Studies and Technology (CAST)

Department of Medical, Oral and Biotechnological Sciences

"G. d'Annunzio" University of Chieti-Pescara (Italy)

---

## [Author Response · Author response to Decision Letter 0]

19 Sep 2019

We copied and pasted here the contents of the uploaded file: "Response to reviewers". We recommend the editor and the reviewers to check that file for a more clear and detailed information. In addition, some figures have been added to the mentioned file to address the questions raised by the reviewers. 

We have now revised our manuscript and file naming aiming at meeting the PLOS ONE's style requirements. We put special emphasis in the correction of author’s affiliations. We however must follow the requirements from our institutions which have very strict policies concerning this point. We hope to find a solution on this point. 

Ok. we decided to remove it as this was not a core part of the study.

"This work was supported by the French "Plan Cancer 2009-2013" (Systems Biology call). The funders had no role in study design, data collection and analysis, decision to publish, or preparation of the manuscript."

 We note that one or more of the authors are employed by a commercial company: 'ProGeLife, Marseille, France'.

We added the sentence: “This work was supported by the French "Plan Cancer 2009-2013" (Systems Biology call). The funders had no role in study design, data collection and analysis, decision to publish, or preparation of the manuscript” to the funding statement and to the cover letter.

We added the sentence: “The funder ProGeLife provided support in the form of salaries for author AV, but had no role in the study design, data collection and analysis, decision to publish, or preparation of the manuscript. The specific roles of the authors are articulated in the ‘author contributions’ section” to the funding statement and to the cover letter.

We added the sentence “This commercial affiliation does not alter our adherence to PLOS ONE policies on sharing data and materials.” to the funding statement and to the cover letter. Please do not hesitate to contact us if we did not understand something correctly.

Reviewer #1: 

There are several studies in the literature about prostate cancer proteomics. However, in some cases these previous publications yielded data with limited reproducibility. It is sometimes difficult to distinguish between data which have implications in the clinic and others. Although data quality is acceptable, there are some questions about the conclusions.

We agree that reproducibility is a major issue in cancer research, and that the translation from observations to clinics is often uncertain. Here are few choices we made to try to mitigate these issues:

We avoided the imputation of missing values in our proteomics datasets, and performed stringent filtering to reduce the impact of technical noise;

- We tried to be particularly cautious in the wording of the manuscript, and to not overstate the clinical implications of the candidates obtained during the exploration of our datasets;

- We selected cell lines widely used as models in prostate cancer research;

- We carefully checked the literature to evaluate if we retrieve (and thereby reproduce) proteins already described associated with prostate cancer.

1. the reason for comparison of sensitive and resistant cell lines are not clear. This could imply that androgen receptor expression decreases during tumor progression what is not the case. It is difficult to draw any conclusion from this comparison.

We are not sure to understand this comment. Indeed, the two castration-resistant cell lines do lack the androgen receptor, and the castration-sensitive cell line expresses it (Pienta et al. 2008, Cunningham et al. 2015). The goal of the comparison of sensitive versus resistant cells was thereby to assess pathways that could be differentially activated in these cell lines, and thereby point towards mechanisms implemented by the resistant cell lines to overcome, among others, the absence of the androgen receptor. Several mechanisms are known to be involved in the emergence of the malignant progression of prostate cancer and resistance phenotype, through AR-related and non-AR-related pathways. For example, post-transcriptional modification using miRNA, epigenetic alterations, alternative splicing or gene fusion are parts of the hallmark of CRPC (Katsogiannou M., Cancer Treat Rev. 2015). We identified several proteins involved in these mechanisms in our proteomics screening. In 30% of CRPC, the AR promoter region has been described to be hypermethylated, resulting in the loss of AR expression in those tumors (Suzuki H, Endocr Relat Cancer 2003).

In the manuscript, we clarified the introduction to state precisely that resistance is associated with androgen receptor pathway changes (line 8 in the introduction). “This progression involves several molecular mechanisms such as ligand-independent androgen receptor activation, androgen receptor (AR) loss or adaptive upregulation of anti-apoptotic genes (for review 3)”.

Pienta KJ, Abate-Shen C, Agus DB, Attar RM, Chung LW, Greenberg NM, Hahn WC, Isaacs JT, Navone NM, Peehl DM, Simons JW, Solit DB, Soule HR, VanDyke TA, Weber MJ, Wu L, Vessella RL. The current state of preclinical prostate cancer animal models. Prostate. 2008May 1;68(6):629-39. doi: 10.1002/pros.20726

CunninghamD, You Z. In vitro and in vivo model systems used in prostate cancer research. J Biol Methods. 2015;2(1). pii: e17.

2. the descriütion of human cell lines in results is not necessary. They are very well known in scientific community.

These prostate cell lines are indeed well-known in the prostate cancer research community. We however expect the dataset we are producing to be used as a resource by other researchers with diverse expertise, such as bioinformatics, and think it’s important to state their biological features in details. We however removed the description of the cell lines from the Material and Methods sections to reduce redundancies.

3. In the Results section, there are too many repetitions of that what has already been presented in Materials and Methods.

We have reviewed the Results and Material & Methods sections in details and removed redundancies. 

4. it is not clear whether this manuscript could confirm previous findings in the proteomic field.

Apart from the proteins for which we manually curated the literature to validate their previous implication in prostate cancer, and that are described in the text, we also considered comparing our results with existing “omics” large-scale datasets. We identified prostate cancer publications focusing on tissue proteomics (Iglesias-Gato et al. 2016) and on cell line & tissue phosphoproteomics (Drake et al. 2016). We also found an update of Iglesias-Gato et al. 2016 on the proteome of prostate cancer bone metastasis (Iglesias-Gato et al. 2018). Finally, genomics data are also available for ~60 cancer cell lines in the NCI60 resource. We discuss below the comparison of our dataset with each of these resources. Please note that this answer is similar to the one proposed for Reviewer 2 first comment.

Iglesias-Gato, D. et al. The Proteome of Primary Prostate Cancer. European Urology 69, 942–952 (2016).

Drake, J. M. et al. Phosphoproteome Integration Reveals Patient-Specific Networks in Prostate Cancer. Cell 166, 1041–1054 (2016).

Iglesias-Gato, D. et al. The Proteome of Prostate Cancer Bone Metastasis Reveals Heterogeneity with Prognostic Implications. Clin. Cancer Res. 24, 5433–5444 (2018).

- Comparison with NCI60 genomics data resource

We checked the NCI-60 tumor cell line screen (https://dtp.cancer.gov/discovery_development/nci-60/), which provide genomics data for a large panel of reference cell lines. Our goal was to try to link genomics mutations in cell lines with proteomics level deregulations. However, only 2 of the 4 cell lines that we profiled in our experiment are described in this database (DU145 and PC3), and we decided not to pursue in this direction.

- Drake et al. (2016)

In Drake et al., the authors integrate transcriptomics and phosphoproteomics (but no proteomics) data for many tumor tissues and 3 prostate cancer cell lines. We studied their paper in-depth, in particular the Figure 1, which shows that the cell lines data were clustering apart from the tissue data. We contacted the authors when the paper was published, and they gave us a more detailed presentation of their Figure 1. Unfortunately, we have only the DU145 cell line in common, and we were thereby not able to compare our clusterings. In addition, a direct comparison of our phosphoproteomics raw data is not pertinent as, contrarily to us, their protocole implies vanadate (a protein-phosphotyrosine phosphatase inhibitor) treatment, in order to increase the detection of the phospho-tyrosine.

- Iglesia-Gato et al. 2016

We also compared the proteomics data we have generated with the ones produced in Iglesias-Gato 2016. We selected the normalised ratios (before imputation of missing values) of the 9829 identified proteins in the Iglesias-Gato dataset. In our dataset, we took the normalised ratios of the 3219 identified proteins. The heatmap below (check Response to reviewers file) was created based on the Pearson correlation values between the expression levels of 2673 proteins identified in both experiments. We can observe that cell lines clusterize apart from tissue data. This clustering is similar to the one obtained by in prostate cancer transcriptomics and phosphoproteomics (Drake et al. 2016) as well as for other cancer types and omics data (Domcke et al. 2013). It has also been observed for breast cancer that among different types of omics, proteins levels had the lowest correlation between tumors and cell lines (Jiang et al. 2016). Overall, these observations highlight the large dissimilarities between tissue samples and cell lines, emphasizing the need to obtain molecular profiles from both types of resources.

Domcke, S., Sinha, R., Levine, D. A., Sander, C. & Schultz, N. Evaluating cell lines as tumour models by comparison of genomic profiles. Nature Communications 4, 2126 (2013).

Drake, J. M. et al. Phosphoproteome Integration Reveals Patient-Specific Networks in Prostate Cancer. Cell 166, 1041–1054 (2016).

Jiang G, Zhang S, Yazdanparast A, et al. Comprehensive comparison of molecular portraits between cell lines and tumors in breast cancer. BMC Genomics. 2016;17 Suppl 7(Suppl 7):525.

We also conducted a meta-analysis using the R package “MetaDE”. Our goal was to compare the protein level trends among tissue and cell lines. We used a Random Effects Model (REM) approach for the meta-analysis, since it allows heterogeneity in the effect sizes between the different datasets (Tissue VS cell lines). We took PNT1A as the reference for our SILAC experiment. Proteins were selected as those displaying an FDR corrected p-value < 0.05. The results are the following:

- LNCaP: 114 proteins have a similar behaviour in cell lines as compared to tumor tissues. This means that the expression level changes of these proteins between LNCaP and PNT1A are similar to those between control tissue and tumorigenic tissue.

- DU145: 81 proteins have a similar behaviour.

- PC3: 73 proteins have a similar behaviour.

These results indicate that LNCaP is the cell line with protein levels displaying the most similar behavior as compared to the tumor tissue. 

Iglesia-gato 2018

We finally applied our previously mentioned comparison protocol to the data produced by Iglesias-Gato in 2018. In this case, they identify 9828 proteins, 2673 of them also match with our identified proteins. Here, we also observe that our protein expression data clusterize separately from the one resulting from tissue samples (See figures in the Response to reviewers file).

Overall, the main strengths of our work are to profile four widely used prostate cell lines and to do this both at the proteomics and phosphoproteomics levels. In addition, as shown in the previous comparisons, our datasets are complementary to the recent literature on prostate cancer transcriptomics, proteomics or phosphoproteomics, and could be useful for the community working with these cell lines in routine. Furthermore, we also propose a bioinformatics exploration of our datasets, as well as their integration into a molecular network, allowing the identification of processes that are not detectable with a traditional analyses of differentially expressed proteins/sites.

Reviewer #2: 

M Katsogiannou et al presented a study that aimed to investigate and explore the proteome and phosho-proteome of four well established prostate cancer cell lines. Utilizing a SILAC-based Mass Spectrometry approach, the data show set of proteins that are commonly and highly expressed in all four cell lines, as well as differentially expressed proteins between castrate-resistant and castrate-sensitive cells. Other comparisons such as proteins up-regulated in cancer cell lines compared to non-tumorigenic cell is demonstrated. Phospho-proteomic data is also presented for the cell lines including presentation of the proteome and phosphor-proteome in a molecular network to identify candidate biomarkers for Prostate cancer. 

Overall, the study is well-done and provides a starting point to further explore the role of potential candidate proteins identified through this study in prostate cancer. Although the study adds some new potential proteins to the list as target molecules and biomarkers, it reconfirms a lot of information that is already published in the field.

We thank the reviewer for this accurate and detailed summary of our manuscript. Many thanks. Indeed, one of the manuscript's main goals is to provide a resource as a starting point for further exploration of proteins related to PC progression. 

1)There are other studies that have used proteomic -based approaches on prostate cancer cells or conditioned media from prostate cancer cells? What is the novelty of the approach used in this study?

Apart from the proteins for which we manually curated the literature to validate their previous implication in prostate cancer, and that are described in the text, we also considered comparing our results with existing “omics” large-scale datasets. Importantly, we did not identify other proteomics/phosphoproteomics approaches focusing on prostate cell lines, but we identified prostate cancer publications focusing on tissue proteomics (Iglesias-Gato et al. 2016) and on cell line & tissue phosphoproteomics (Drake et al. 2016). We also found an update of Iglesias-Gato et al. 2016 on the proteome of prostate cancer bone metastasis (Iglesias-Gato et al. 2018). Finally, genomics data are also available for ~60 cancer cell lines in the NCI60 resource. We discuss below the comparison of our dataset with each of these resources. Please note that this answer is similar to the one proposed for Reviewer 1 fourth comment.

Iglesias-Gato, D. et al. The Proteome of Primary Prostate Cancer. European Urology 69, 942–952 (2016).

Drake, J. M. et al. Phosphoproteome Integration Reveals Patient-Specific Networks in Prostate Cancer. Cell 166, 1041–1054 (2016).

Iglesias-Gato, D. et al. The Proteome of Prostate Cancer Bone Metastasis Reveals Heterogeneity with Prognostic Implications. Clin. Cancer Res. 24, 5433–5444 (2018).

- Comparison with NCI60 genomics data resource

We checked the NCI-60 tumor cell line screen (https://dtp.cancer.gov/discovery_development/nci-60/), which provide genomics data for a large panel of reference cell lines. Our goal was to try linking genomics mutations in cell lines with proteomics level deregulations. However, only 2 of the 4 cell lines that we profiled in our experiment are described in this database (DU145 and PC3), and we decided not to pursue in this direction.

- Drake et al. (2016)

In Drake et al., the authors integrate transcriptomics and phosphoproteomics (but no proteomics) data for many tumor tissues and 3 prostate cancer cell lines. We studied their paper in-depth, in particular the Figure 1, which shows that the cell lines data were clustering apart from the tissue data. We contacted the authors when the paper was published, and they gave us a more detailed presentation of their Figure 1. Unfortunately, we have only the DU145 cell line in common, and we were thereby not able to compare our clusterings. In addition, a direct comparison of our phosphoproteomics raw data is not pertinent as, contrarily to us, their protocole implies vanadate (a protein-phosphotyrosine phosphatase inhibitor) treatment, in order to increase the detection of the phospho-tyrosine.

- Iglesia-Gato et al. 2016

We also compared the proteomics data we have generated with the ones produced in Iglesias-Gato 2016. We selected the normalised ratios (before imputation of missing values) of the 9829 identified proteins in the Iglesias-Gato dataset. In our dataset, we took the normalised ratios of the 3219 identified proteins. The heatmap below (check response to reviewers file) )was created based on the pearson correlation values between the expression levels of 2673 proteins identified in both experiments. We can observe that cell lines clusterize apart from tissue data. This clustering is similar to the one obtained by in prostate cancer transcriptomics and phosphoproteomics (Drake et al. 2016) as well as for other cancer types and omics data (Domcke et al. 2013). It has also been observed for breast cancer that among different types of omics, proteins levels had the lowest correlation between tumors and cell lines (Jiang et al. 2016). Overall, these observations highlight the large dissimilarities between tissue samples and cell lines, emphasizing the need to obtain molecular profiles from both types of resources.

Domcke, S., Sinha, R., Levine, D. A., Sander, C. & Schultz, N. Evaluating cell lines as tumour models by comparison of genomic profiles. Nature Communications 4, 2126 (2013).

Drake, J. M. et al. Phosphoproteome Integration Reveals Patient-Specific Networks in Prostate Cancer. Cell 166, 1041–1054 (2016).

Jiang G, Zhang S, Yazdanparast A, et al. Comprehensive comparison of molecular portraits between cell lines and tumors in breast cancer. BMC Genomics. 2016;17 Suppl 7(Suppl 7):525.

We also conducted a meta-analysis using the R package “MetaDE”. Our goal was to compare the protein level trends among tissue and cell lines. We used a Random Effects Model (REM) approach for the meta-analysis, since it allows heterogeneity in the effect sizes between the different datasets (Tissue VS cell lines). We took PNT1A as the reference for our SILAC experiment. Proteins were selected as those displaying an FDR corrected p-value < 0.05. The results are the following:

- LNCaP: 114 proteins have a similar behaviour in cell lines as compared to tumor tissues. This means that the expression level changes of these proteins between LNCaP and PNT1A are similar to those between control tissue and tumorigenic tissue.

- DU145: 81 proteins have a similar behaviour.

- PC3: 73 proteins have a similar behaviour.

These results indicate that LNCaP is the cell line with protein levels displaying the most similar behavior as compared to the tumor tissue. 

Iglesia-gato 2018

We finally applied our previously mentioned comparison protocol to the data produced by Iglesias-Gato in 2018. In this case, they identify 9828 proteins, 2673 of them also match with our identified proteins. Here, we also observe that our protein expression data clusterize separately from the one resulting from tissue samples (check figures in Response to reviewers file).

Overall, the main strengths and novelty of our work are to profile four widely used prostate cell lines and to do this both at the proteomics and phosphoproteomics levels. As shown in the previous comparisons, our datasets are complementary to the recent literature on prostate cancer transcriptomics, proteomics or phosphoproteomics, and could be useful for the community working with these cell lines in routine. Furthermore, we also propose a bioinformatics exploration of our datasets, as well as their integration into a molecular network, allowing the identification of processes that are not detectable with a traditional analyses of differentially expressed proteins/sites.

2) Please show if the data can be used for miRNA target prediction? It would be a good addition to the study as miRNA’s have great potential as biomarkers in cancer.

This is indeed a very interesting point. During our review of the literature, we explored some interesting cases of miRNA-target interactions that could be relevant in the context of our prostate proteomics study (Vanacore et al. 2017). For instance, the regulation of VDAC1 by miR-29 is very appealing. VDAC1 is a mitochondrial protein directly involved in apoptosis (it promotes the apoptosis of tumor cells), and it harbours a miR-29a target site. The overexpression of miR-29a has been shown to result in downregulation of VDAC1 (Bargaje et al. 2012). The protein level profile of VDAC1 in the different prostate cells lines is shown below (check Response to reviewers file). Low levels in the expression of VDAC1 could lead to a decrease in the apoptosis of tumor cells resulting in more aggressive cancers as represented by PC3 and DU145 lines. miR-29 might be involved in this process, it would be interesting to profile its expression in these cell lines. We however acknowledge that we did not design our study to consider miRNA-target predictions. For instance, the network we are building considers proteins and phosphosites measured in our study, but we did not measured miRNA. It would be an attractive follow-up.

Vanacore D et al. Micrornas in prostate cancer: an overview. Oncotarget. 2017 Jul 25;8(30):50240-50251.

Bargaje R et al. Identification of novel targets for miR-29a using miRNA proteomics. PLoS One. 2012;7(8):e43243. doi:10.1371/journal.pone.0043243

3) Add PCA plot to show that each cell line has a unique protein and phosphor-protein signature and add Venn-diagrams to show common and unique expressors for a) cancer cell lines compared to non-tumorigenic cells b) castrate resistant cells compared to castrate sensitive cells.

Please, check the plots and comments in the Response to Reviewers file.

4) Are there common expressors and unique expressors between DU145 and PC3. This could perhaps help identify candidate biomarkers and targets for highly metastatic and aggressive disease.

This is a very interesting comparison. We performed t-test comparisons between the two castration-resistant cell lines to determine differentially expressed proteins and phosphosites. We finally did not presented this information because we first aimed at exploring the molecular events driving the global progression of prostate cancer to resistance, i.e., we focused on the mechanisms that could be common in the two resistant cell lines as compared to the sensitive one. However, global similarities and differences between PC3 and DU145 can be appraised in the context of their comparison to PNT1A and LNCaP in the manuscript. 

5) Show box-plots for expression levels of TAGLN2 and HNRNPN1 in an additional figure. Validation of differences in protein levels must be shown.

We assumed that the reviewer refers to TAGLN2 and HNRNPA1. These two proteins are interesting candidates because they have been associated with phosphosites significantly different from one cell line to another according to an ANOVA test. We are not sure to understand the comment on the validation of differences in protein levels. Both proteins are housekeeping (top 10% of most expressed proteins in all cell lines according to iBAQ values). All proteins discussed in the manuscript have been identified in the proteomics experiment with an FDR < 1%. TAGLN2 is in addition under-expressed in LNCaP cell line (FDR=0.007), and over-expressed in DU145 (FDR=0.02) and PC3 (FDR=0.049) cell lines, as compared with PNT1A, and over-expressed in resistant compared to sensitive cell lines (FDR=0.01). As suggested by the reviewer, box-plots describing the protein and phosphosite levels of TAGLN2 and HNRNPA1 have been generated and are included in the submission of the new version of the manuscript as supplementary figures 4 and 5. 

5) The resolution is images in the figures needs to be increased. Figures are hard to read as of now.

We increased the resolution the final version of the manuscript. In addition, we increased the size and quality of the labels in figures 2 and 3. 

Reviewer #3: 

The manuscript from Katsogiannou et al. showed a large SILAC-based Mass Spectrometry experiment that allowed to map the proteomes and phosphoproteomes of PNT1A, LNCaP, DU145 and PC3 prostate cancer cell lines, and reveal different signaling networks associated with the cellular context of each cell line, possibly reflecting the pathological features of human Prostate Cancer (hormonal status, ability to metastatize etc.).

The experimental data are strong, rigorous and well presented, in particular the deep comparison of the four cell lines for the identification of the housekeeping proteome vs the most significant variations across the samples.

We thank the reviewer for his encouraging comments and for the critical points discussed, which help us to improve the quality of our manuscript. 

However, some critical points emerged, and should be clarified/investigated more in depth:

1. The androgen-dependent/castration-sensitive cell line LNCaP has been compared with the androgen-independent/castration-resistant cells DU145 and PC3. However, the best in vitro models for comparing these two PrCa conditions would be LNCaP vs C4-2. Why the authors did exclude C4-2 cells from their high-throughput analysis?

Several other PCa models exist, such as C4-2, a particularly interesting LNCaP-derived cell line that constitutes a compelling androgen-independent model. We however had to make a choice in the number of cell lines during the initial design of the study, because of the amount of experimental material and subsequent data produced. We decided not to work with derived cell lines (from LNCaP or other cells), because of the genetic proximity. We wanted instead to identify mechanisms of resistance that could have emerged in genetically different cell lines, derived from separate metastasis. The cells lines described in our study are four of the most widely used cell lines in prostate cancer research (Pienta et al. 2008, Cunningham et al. 2015) that we routinely use in the lab. 

Pienta KJ, Abate-Shen C, Agus DB, Attar RM, Chung LW, Greenberg NM, Hahn WC, Isaacs JT, Navone NM, Peehl DM, Simons JW, Solit DB, Soule HR, VanDyke TA, Weber MJ, Wu L, Vessella RL. The current state of preclinical prostate cancer animal models. Prostate. 2008May 1;68(6):629-39. doi: 10.1002/pros.20726

CunninghamD, You Z. In vitro and in vivo model systems used in prostate cancer research. J Biol Methods. 2015;2(1). pii: e17.

2. The ANOVA analysis of the proteomics/phosphoproteomics data highlighted several proteins/phosphosites that vary significantly in various comparisons (benign vs malignant, CR vs CS etc.). For some of these variations, the authors even claimed that “these proteins could constitute markers of oncogenic transformation”. To support this kind of statements, the authors should provide “wet-lab” validations of their high-throughput results, at least on representative targets among those described in the text (e.g. Septin-9, TAGLN2, HNRNPA1, RAB5B/RAB7A, TriC/CCT complex, TP53BP1 pSer-500 and pThr-1056, DDX10 pSer-539 etc). This type of validation would also help the authors focusing on the most important pathways, rather than leaving the reader with a comprehensive description of all signaling networks potentially involved in the regulation of PrCa malignant progression.

The generation of the SILAC proteomics and phosphoproteomics dataset from the 4 cell lines and with 3 replicates was a large and complex experimental work, and we decided to focus our subsequent analyses on the in-depth data exploration of this dataset rather than on functional experimental validation of cherry-picked candidates. In the design of our study, we took inspiration from references in the proteomics field, in particular from Matthias Mann lab papers on cell line proteomics characterizations (Lundberg et al. 2010, Geiger et al. 2012), but also from a reference in prostate cancer analysis in which the authors produced phosphoproteomics and genomics for prostate cancer, both on cell lines and real tumor tissue (Drake et al. 2016). Similarly to these studies, we concentrated our efforts on the data generation and their exploration, with the objective to produce a resource that could subsequently be used as a starting point for further hypothesis and experimental investigations (by us and others, as, importantly, we make all data available [ProteomeXchange Consortium (www.proteomexchange.org) via the PRIDE partner repository with datasets identifiers PXD004970 and PXD004992.]).

Lundberg, E. et al. Defining the transcriptome and proteome in three functionally different human cell lines. Molecular Systems Biology 6, (2010).

Geiger, T., Wehner, A., Schaab, C., Cox, J. & Mann, M. Comparative Proteomic Analysis of Eleven Common Cell Lines Reveals Ubiquitous but Varying Expression of Most Proteins. Mol Cell Proteomics 11, (2012).

Drake, J. M. et al. Phosphoproteome Integration Reveals Patient-Specific Networks in Prostate Cancer. Cell 166, 1041–1054 (2016).

Minor points:

1. Page 8, Lanes 303-305): DU145 are derived from CNS metastasis, and PC3 from bone metastasis. Please revise the sentence.

Thanks for catching this mistake! This has been corrected.

2. Figure 2: it should be useful to include a “title/legend” to each bar graph (e.g. LNCaP vs PNT1A in the panel A), showing the comparisons as described in the figure legend text.

Done

3. Figure 3: similar to the previous point, it should be useful to include a “visible” title/legend to each panel (e.g. DU145/PC3 vs LNCaP in the panel A, and CR vs CS on top of the panel B).

Done. We have now included a title for each subfigure to further clarify the different plots as suggested. Moreover, we increased the size of the label titles and numbers in the axis. Done

---

## [Decision Letter · Decision Letter 1]

8 Oct 2019

Integrative proteomic and phosphoproteomic profiling of prostate cell lines

PONE-D-19-20799R1

Dear Dr. Valdeolivas,

We are pleased to inform you that your manuscript has been judged scientifically suitable for publication and will be formally accepted for publication once it complies with all outstanding technical requirements.

With kind regards,

Lucia R. Languino, Ph.D.

Academic Editor

PLOS ONE

Additional Editor Comments (optional):

Reviewers' comments:

Reviewer's Responses to Questions

**Comments to the Author**

1. If the authors have adequately addressed your comments raised in a previous round of review and you feel that this manuscript is now acceptable for publication, you may indicate that here to bypass the “Comments to the Author” section, enter your conflict of interest statement in the “Confidential to Editor” section, and submit your "Accept" recommendation.

Reviewer #1: All comments have been addressed

Reviewer #2: All comments have been addressed

Reviewer #3: All comments have been addressed

2. Is the manuscript technically sound, and do the data support the conclusions?

Reviewer #1: Yes

Reviewer #2: Yes

Reviewer #3: Yes

3. Has the statistical analysis been performed appropriately and rigorously? 

Reviewer #1: Yes

Reviewer #2: Yes

Reviewer #3: Yes

4. Have the authors made all data underlying the findings in their manuscript fully available?

Reviewer #1: Yes

Reviewer #2: Yes

Reviewer #3: Yes

5. Is the manuscript presented in an intelligible fashion and written in standard English?

Reviewer #1: Yes

Reviewer #2: Yes

Reviewer #3: Yes

6. Review Comments to the Author

Reviewer #1: (No Response)

Reviewer #2: The authors have added data as requested, which helps tie the pieces of the manuscript together in a better manner. Changes have been made to the original figures which add clarity and readability of the manuscript and enhance the overall impact of the study. Explanations provided in response to reviewer comments are decent as authors provide a convincing response regarding the reasons behind the questions they asked in the study. Overall, the manuscript is well- done and provides a good starting point to further explore the potential candidate proteins and conduct follow-up studies such as miRNA targets etc which the authors of the study themselves appreciate as a strength of the study.

Reviewer #3: (No Response)

7. PLOS authors have the option to publish the peer review history of their article (what does this mean?). If published, this will include your full peer review and any attached files.

Reviewer #1: No

Reviewer #2: No

Reviewer #3: Yes: Marco Trerotola

---

## [Editor Report · Acceptance letter]

21 Oct 2019

PONE-D-19-20799R1 

Integrative proteomic and phosphoproteomic profiling of prostate cell lines 

Dear Dr. Valdeolivas:

I am pleased to inform you that your manuscript has been deemed suitable for publication in PLOS ONE. Congratulations! Your manuscript is now with our production department. 

With kind regards,

on behalf of

Dr. Lucia R. Languino 

Academic Editor

PLOS ONE